# Histone acetylome-wide associations in immune cells from individuals with active *Mycobacterium tuberculosis* infection

Ricardo C. H. del Rosario [1,2,3,16], Jeremie Poschmann [1,4,16], Carey Lim[5,6,17], Catherine Y. Cheng[5,17], Pavanish Kumar [5], Catherine Riou [7,8,9], Seow Theng Ong [10], Sherif Gerges[2,3,11], Hajira Shreen Hajan[1], Dilip Kumar [4], Mardiana Marzuki[5,6], Xiaohua Lu[5], Andrea Lee[5,6], Giovani Claresta Wijaya[1], Nirmala Arul Rayan[1], Zhong Zhuang[10], Elsa Du Bruyn[7,8], Cynthia Bin Eng Chee[12], Bernett Lee[5], Josephine Lum [5], Francesca Zolezzi[5], Michael Poidinger [5], Olaf Rotzschke [5], Chiea Chuen Khor [1], Robert J. Wilkinson[7,8,13,14], Yee T. Wang[12], George K Chandy[10], Gennaro De Libero[5,15], Amit Singhal[5,6,10 ✉] and Shyam Prabhakar [1 ✉]

**Host cell chromatin changes are thought to play an important role in the pathogenesis of infectious diseases. Here we describe a histone acetylome-wide association study (HAWAS) of an infectious disease, on the basis of genome-wide H3K27 acetylation profiling of peripheral blood granulocytes and monocytes from persons with active *Mycobacterium tuberculosis* (*Mtb*) infection and healthy controls. We detected >2,000 differentially acetylated loci in either cell type in a Singapore Chinese discovery cohort (n = 46), which were validated in a subsequent multi-ethnic Singapore cohort (n = 29), as well as a longitudinal cohort from South Africa (n = 26), thus demonstrating that HAWAS can be independently corroborated. Acetylation changes were correlated with differential gene expression. Differential acetylation was enriched near potassium channel genes, including *KCNJ15*, which modulates apoptosis and promotes *Mtb* clearance in vitro. We performed histone acetylation quantitative trait locus (haQTL) analysis on the dataset and identified 69 candidate causal variants for immune phenotypes among granulocyte haQTLs and 83 among monocyte haQTLs. Our study provides proof-of-principle for HAWAS to infer mechanisms of host response to pathogens.**

Tuberculosis (TB), caused by the bacterium *Mycobacterium tuberculosis* (*Mtb*), results in 1.5 million deaths per year[1]; it is thus the second largest cause of mortality among infectious diseases globally. *Mtb* circumvents the immune system by remodelling the host transcriptome, leading to inhibition of protective and induction of pathological immune responses[2–4]. In particular, gene expression studies on whole blood from active TB (ATB) patients suggest that alterations in pathways such as interferon signalling, inflammation, apoptosis and pattern recognition receptor signalling may contribute to TB pathogenesis[4–6]. It is possible that these transcriptomic changes in host immune cells during infection are chromatin-mediated[7–10].

Histone H3 acetylation at lysine 27 (H3K27ac) is a well-established chromatin signature of active enhancers and promoters[11,12] that correlates with gene expression and transcription factor binding[13]. We previously profiled H3K27ac genome-wide in post-mortem autism spectrum disorder (ASD) brain vs control, and detected widespread ASD-associated chromatin perturbations converging upon specific pathways, thus providing a resource for subsequent mechanistic studies of ASD pathology[14]. Two other recent studies used the same methodology to detect thousands of histone acetylation changes in post-mortem brain samples from individuals with Alzheimer's disease[15,16], and more recent studies examined acetylation changes in dilated cardiomyopathy[17] and heart failure[18]. We hypothesize that this histone acetylome-wide association study (HAWAS) approach could also provide molecular insights into non-neurological conditions, such as infectious diseases.

Here we report a HAWAS of peripheral blood monocytes and granulocytes from ATB patients and age-matched healthy controls. Uniquely, we profiled histone acetylation in two distinct cohorts from Singapore (discovery and validation), as well as a longitudinal cohort from South Africa that included both ATB and latent

[1]Genome Institute of Singapore, Agency for Science, Technology and Research (A*STAR), Singapore, Singapore. [2]Stanley Center for Psychiatric Research, Broad Institute of MIT and Harvard, Cambridge, MA, USA. [3]Department of Genetics, Harvard Medical School, Boston, MA, USA. [4]Inserm, Université de Nantes, Centre de Recherche en Transplantation et Immunologie, ITUN, Nantes, France. [5]Singapore Immunology Network, A*STAR, Singapore, Singapore. [6]A*STAR Infectious Diseases Labs, A*STAR, Singapore, Singapore. [7]Wellcome Centre for Infectious Diseases Research in Africa, University of Cape Town, Cape Town, South Africa. [8]Institute of Infectious Disease and Molecular Medicine, University of Cape Town, Cape Town, South Africa. [9]Division of Medical Virology, Department of Pathology, University of Cape Town, Cape Town, South Africa. [10]Lee Kong Chian School of Medicine, Nanyang Technological University, Singapore, Singapore. [11]Analytic and Translational Genetics Unit, Department of Medicine, Massachusetts General Hospital, Boston, MA, USA. [12]Tuberculosis Control Unit, Tan Tock Seng Hospital, Singapore, Singapore. [13]Department of Infectious Disease, Imperial College, London, UK. [14]The Francis Crick Institute, London, UK. [15]Department of Biomedicine, University of Basel, Basel, Switzerland. [16]These authors contributed equally: Ricardo C. H. del Rosario, Jeremie Poschmann. [17]These authors contributed equally: Carey Lim, Catherine Y. Cheng. ✉e-mail: amit_singhal@idlabs.a-star.edu.sg; prabhakars@gis.a-star.edu.sg

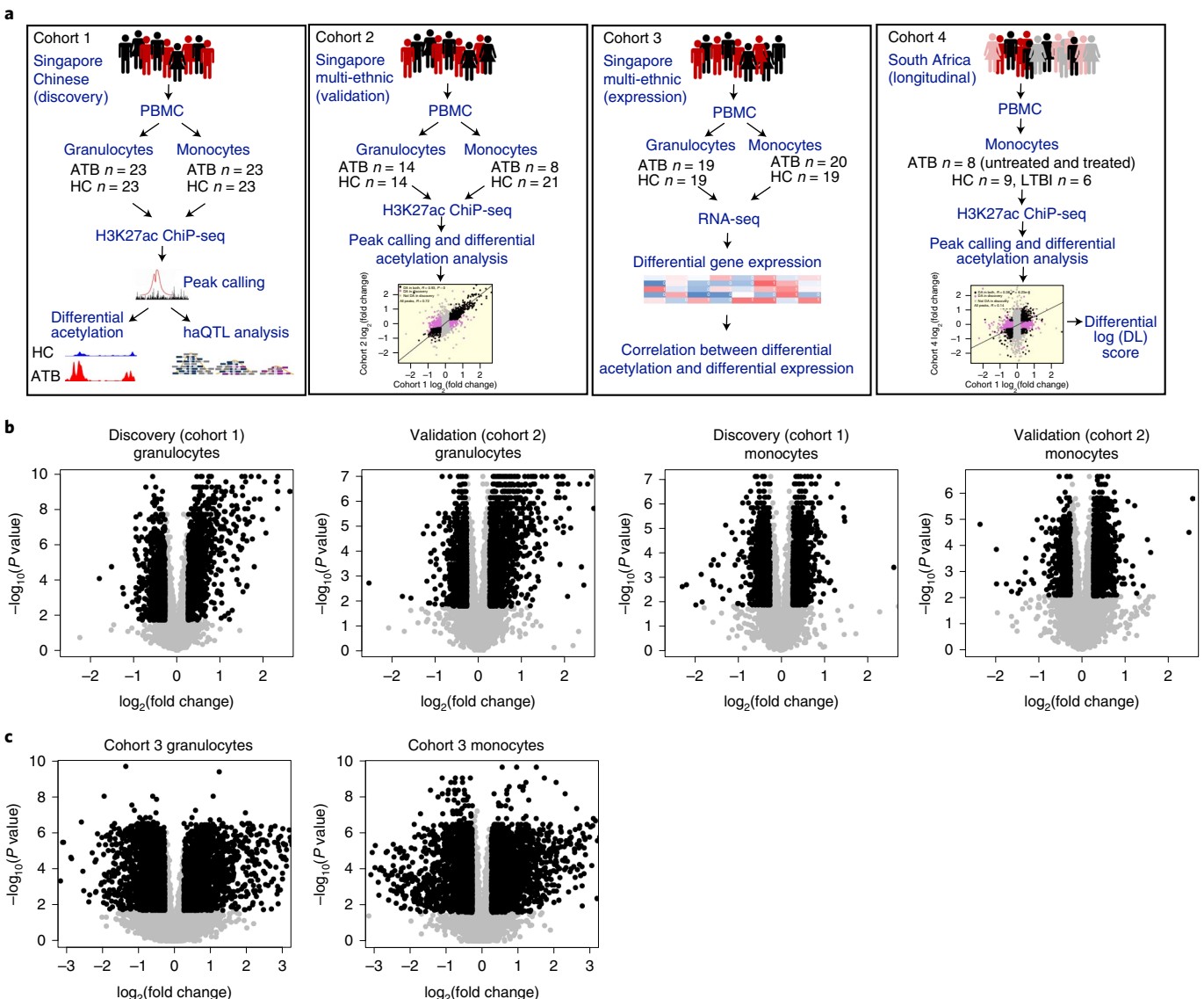

**Fig. 1 | Histone acetylome-wide association study of TB. a**, Overview of data generation and analysis. **b**, Volcano plots showing log$_2$(fold change) and *P* value for consensus ChIP-seq peaks (black dots, DA peaks; see Methods). *P* value for each peak was calculated using a 2-sample two-sided Wilcoxon test and fold change was computed from the median of the peak heights. **c**, Volcano plots showing log$_2$(fold change) and *P* value for expressed genes (black dots, DE genes, grey dots, non-DE genes; see Methods). *P* value for each gene was calculated using a 2-sample two-sided Wilcoxon test and fold change was computed from the median of the gene expression.

TB infection (LTBI). We also generated transcriptome data from an additional cohort in Singapore for further corroboration of infection-associated chromatin changes. We then analysed the data to infer specific host-response mechanisms. Finally, we used the chromatin immunoprecipitation with massively parallel DNA sequencing (ChIP-seq) data to call SNPs within monocyte and granulocyte regulatory elements and identify histone acetylation quantitative trait loci (haQTLs), some of which may contribute to inflammatory and infectious disease susceptibility.

## Results

**HAWAS of active tuberculosis.** We used ChIP-seq to profile H3K27ac in a total of 190 peripheral blood granulocyte and monocyte samples from ATB patients and age-matched healthy controls (HC) in Singapore (Fig. 1a), of which 135 were retained after quality control (QC; Supplementary Tables 1–13). These 135 samples from Singapore were split into discovery (Cohort 1,

exclusively Chinese ethnicity) and validation (Cohort 2, multi-ethnic: Chinese, Malay, Indian) cohorts (Fig. 1a), and regulatory elements in each cohort were detected as focal peaks in the ChIP-seq signal (Methods). Our granulocyte and monocyte H3K27ac profiles were highly consistent with those from the International Human Epigenome Consortium[19] (IHEC; Supplementary Fig. 1) and peak heights were strongly correlated (granulocytes $R = 0.90$; monocytes $R = 0.87$; Methods).

To identify infection-associated changes in the regulatory elements, we tested each of them for differential histone acetylation (differential peak height) between ATB and HC samples in the two Singapore cohorts after controlling for confounders (Extended Data Fig. 1, Supplementary Fig. 2 and Methods). In total, we identified >1,800 differentially acetylated (DA) peaks in each combination of cell type and cohort (Fig. 1b and Supplementary Tables 1–13), indicating that ATB-associated chromatin changes were widespread in both granulocytes and monocytes. Moreover, the observed histone

acetylation changes were shared across ATB individuals, rather than limited to a subset (Extended Data Fig. 2a).

Next we investigated the correspondence between histone acetylation changes and differential gene expression. Note that, although absolute histone acetylation and gene expression readouts are strongly correlated across a single genome[13], we expect measures of differential acetylation and differential expression to show only moderate correlation, largely due to lower signal-to-noise ratio[14]. We performed RNA-seq on peripheral blood granulocytes and monocytes from an independent Singapore ATB vs HC cohort from multiple ethnicities (Cohort 3, Fig. 1a) and identified 1,800 differentially expressed (DE) genes in either cell type (Fig. 1c, Supplementary Fig. 3 and Tables 14–17, and Methods). As expected, DE-gene fold changes in granulocytes and monocytes were significantly correlated with fold changes of flanking DA peaks (Supplementary Table 18 and Extended Data Fig. 2b,c), and thus the two data types corroborated each other. However, the correlation was only moderate (granulocytes $R = 0.71$; monocytes $R = 0.30$), as was also observed in our previous HAWAS study of ASD[14] ($R = 0.33–0.38$). These results further support our previous finding that, while differential H3K27ac ChIP-seq and transcriptome profiles tend to correlate globally, they are not mutually redundant. Rather, they provide complementary views of the disease-associated changes under investigation.

One issue not addressed in previous HAWAS studies[14–16] is reproducibility of DA signals. First, we inspected a cluster of DA peaks detected in both granulocytes and monocytes from the discovery cohort near the Guanylate Binding Protein (*GBP*) genes, whose expression has previously been associated with TB susceptibility and host response[10,20]. Almost all of these peaks were also detected as DA in the validation cohort (Fig. 2a). Genome-wide, ATB vs HC peak height fold changes were highly correlated between the Singapore granulocyte discovery and validation cohorts ($R = 0.93$ for DA peaks; Fig. 2b). Moreover, fold-change direction was also highly concordant (99% for DA peaks; $P < 1 \times 10^{-300}$, Fisher's exact test). Peak height fold changes were less correlated in monocytes ($R = 0.70$ for DA peaks; Fig. 2b), most probably due to the smaller number of monocyte ATB samples in the validation cohort (Supplementary Table 1). Nevertheless, the concordance of fold-change direction was still highly significant (96% for DA peaks, $P = 5.2 \times 10^{-61}$, Fisher's exact test), thus confirming the robustness of DA signals detected in the discovery cohort.

To evaluate the generality of ATB-associated chromatin changes identified in the exclusively Chinese discovery cohort, we split the validation cohort into Chinese, Malay and Indian subgroups. The smaller size of these subgroups tended to decrease the accuracy of ATB vs HC fold-change estimates, and thus reduced the correlation between discovery and validation. Nevertheless, for granulocytes, the correlation between discovery and each of the three validation subgroups remained high ($R = 0.82–0.88$; Extended Data Fig. 2d). Monocytes showed a similar trend, except for the Chinese monocyte validation dataset, which contained only one ATB individual (Extended Data Fig. 2d). Thus, the global host chromatin response detected in the exclusively Chinese discovery cohort was also shared by Chinese, Malay and Indian participants from the validation cohort.

To evaluate overall chromatin divergence between ATB and HC, we performed principal component analysis (PCA) on discovery DA peaks (Fig. 2c). As expected, the first principal component (PC1) separated ATB and HC samples in both granulocytes ($P = 4.5 \times 10^{-9}$; two-sided *t*-test) and monocytes ($P = 1.3 \times 10^{-5}$). We then used discovery DA peaks to evaluate chromatin divergence between ATB and HC in the validation cohort. Notably, PC1 based on discovery DA peaks separated ATB from HC in the validation cohort (Fig. 2d; granulocytes $P = 3.1 \times 10^{-6}$; monocytes $P = 9.7 \times 10^{-4}$), further underscoring the validity of DA peaks detected in the discovery

cohort. Our results were qualitatively consistent even when all histone acetylation peaks were analysed, as opposed to only DA peaks (Supplementary Fig. 4). We did not detect any significant effect of smoking status, diabetes and alcohol consumption on PC1 scores (Supplementary Fig. 5). In summary, the above results indicate that over 1,800 histone acetylation foci are altered in circulating granulocytes and monocytes in response to *Mtb* infection.

**Statistical power of HAWAS.** The reproducibility of DA peaks between discovery and validation cohorts suggested that the statistical power of HAWAS was adequate even at relatively small cohort sizes. We therefore generated simulated H3K27ac ChIP-seq datasets by resampling and scaling peak heights from the discovery cohort, and then used these simulated datasets to estimate the power (sensitivity) of DA peak analysis as a function of cohort size and acetylation fold change (Methods). At the fold-change cut-off used in our study ($\geq 1.2$), given the size of the discovery cohort for granulocytes (46) and monocytes (32), this analysis yields power estimates of ~92% and ~81%, respectively (Supplementary Fig. 6). This result from power analysis is consistent with our empirical observation that HAWAS is adequately powered at the cohort sizes used in this study.

**Reproducibility of chromatin changes in South African cohort.** We next investigated the reproducibility of DA peaks in a fourth, geographically distinct cohort from South Africa (SA cohort, Cohort 4, Fig. 1a and Supplementary Table 19). To distinguish between chromatin signatures of active and latent *Mtb* infection, this cohort included both ATB and LTBI individuals. Lastly, to characterize response to drug treatment, we included ATB individuals at baseline (ATB-BL) and after 24 weeks of treatment (ATB-W24). Since we only had access to peripheral blood mononuclear cells (PBMCs) from this cohort, we purified monocytes from each sample and performed H3K27ac ChIP-seq as before. The number of high-quality datasets after QC are 4 ATB-BL, 7 ATB-W24, 9 HC and 6 LTBI. Despite ethnic, geographical and lab-specific differences (PBMC isolation), the majority of DA peaks from the discovery cohort from Singapore were reproduced in the SA cohort ($R = 0.38$ for DA peaks; Fig. 3a–c). Moreover, fold-change direction was also concordant (86% for DA peaks; $P = 6.2 \times 10^{-8}$, Fisher's exact test).

To investigate the effect of LTBI and TB drug treatment on histone acetylation genome-wide, we calculated, for each sample, the average log-transformed histone acetylation signal at ChIP-seq peaks upregulated in the discovery cohort (Up DA peaks). As expected, this signal was elevated among baseline untreated ATB samples relative to HC ($P = 0.0061$, *t*-test; Fig. 3d). In contrast, the signal of LTBI samples was almost identical to that of HC samples, indicating that the increase in histone acetylation was specific to active *Mtb* infection. Moreover, after 24 weeks of treatment, the acetylation signal in ATB individuals dropped to levels similar to HC, indicating that host chromatin response gradually receded as the infection was cleared.

**Enriched pathways and gene loci.** We used the genomic regions enrichment of annotations tool (GREAT)[21] to identify pathways and gene categories significantly altered in response to *Mtb* infection, on the basis of DA peak enrichment at the corresponding gene loci. Upregulated granulocyte DA peaks were enriched for ontologies such as defence response to virus and innate immune response (Fig. 4a and Supplementary Tables 20 and 21). These categories contained several type I interferon-induced genes from the *IFIT* and *IFITM* families, regulators of type I interferon response (*STAT1*, *STAT2*), as well as interferon response factors (*IRFs*; Supplementary Tables 20 and 21). Type I interferon signalling genes were also enriched, although at a lower level, near monocyte upregulated DA peaks.

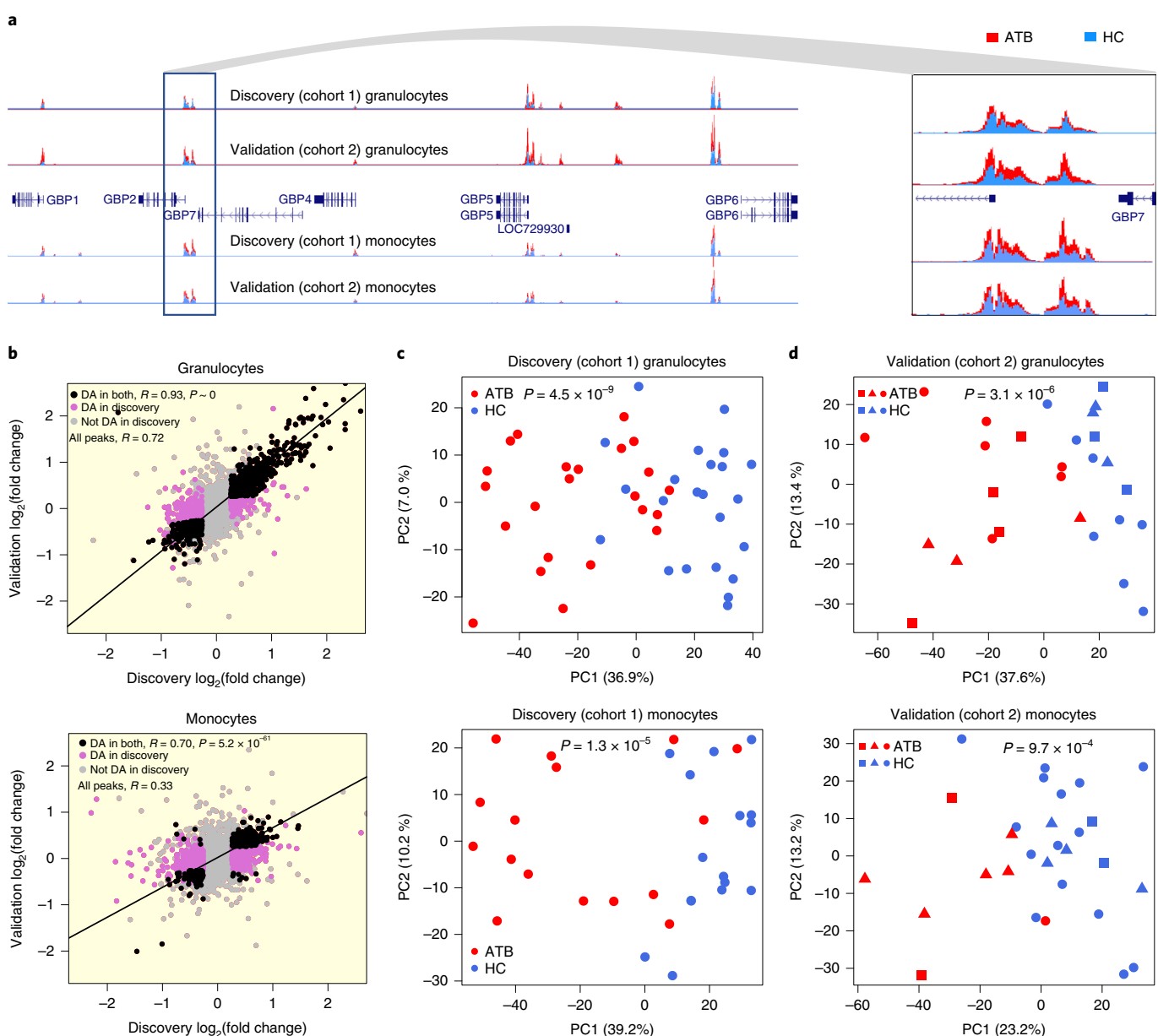

**Fig. 2 | Differentially acetylated (DA) peaks between ATB and HC. a**, Histone acetylation profiles showing consistent differences between ATB (red) and HC (blue) in the *GBP* locus. **b**, Scatterplot of ATB-vs-HC fold change in Singapore discovery and validation cohorts. *R* indicates Pearson correlation coefficient of log$_2$(fold change) between discovery and validation. *P* value of concordance in fold-change direction for shared DA peaks was calculated using two-sided Fisher's exact test. Black dots are DA both in discovery and validation cohorts. Pink dots are DA only in the discovery cohort. Grey dots are not DA in discovery cohort. **c**, Discovery cohort (exclusively Chinese). PCA of granulocyte and monocyte peak heights. **d**, Validation cohort (multi-ethnic). PCA of granulocyte and monocyte peak heights. In **c** and **d**, PCA was based on the discovery DA peak set and *P* values are from two-sided *t*-test of ATB-vs-HC PC1 values. Squares, Indian; circles, Chinese; triangles, Malay.

Upregulated monocyte DA peaks were most prominently enriched near chromatin assembly genes, indicating widespread chromatin reorganization in monocytes in response to TB (Fig. 4a and Supplementary Tables 22 and 23). Notably, genes related to voltage-gated potassium channel activity were also enriched ($P = 9.8 \times 10^{-5}$). Most prominently, the inwardly rectifying potassium channel subfamily J member 15 (*KCNJ15*) gene locus contained 8 upregulated peaks in monocytes, suggesting extensive chromatin changes upon *Mtb* infection (Fig. 4b). This locus also contained 5 upregulated peaks in granulocytes.

To better understand the mechanisms leading to chromatin changes, we investigated the enrichment of transcription factor

(TF) binding motifs in the DA peaks (Methods). We found that both granulocyte and monocyte upregulated peaks contained an excess of STAT and IRF-family motifs, whereas downregulated peaks were enriched for uclear factor kappa B (NF-κB) motifs (Supplementary Tables 24–28). To investigate functional associations of TF binding sites, we further examined DA peaks within each enriched Gene Ontology category (Fig. 4a and Methods). Interestingly, the above-mentioned TF motifs, in addition to motifs from the erythroblast transformation specifi (ETS) and activator protein (AP1) families, associated with distinct functional terms, suggesting that they may regulate distinct functional changes in response to TB. While the enriched motifs from the STAT, IRF and AP1 families

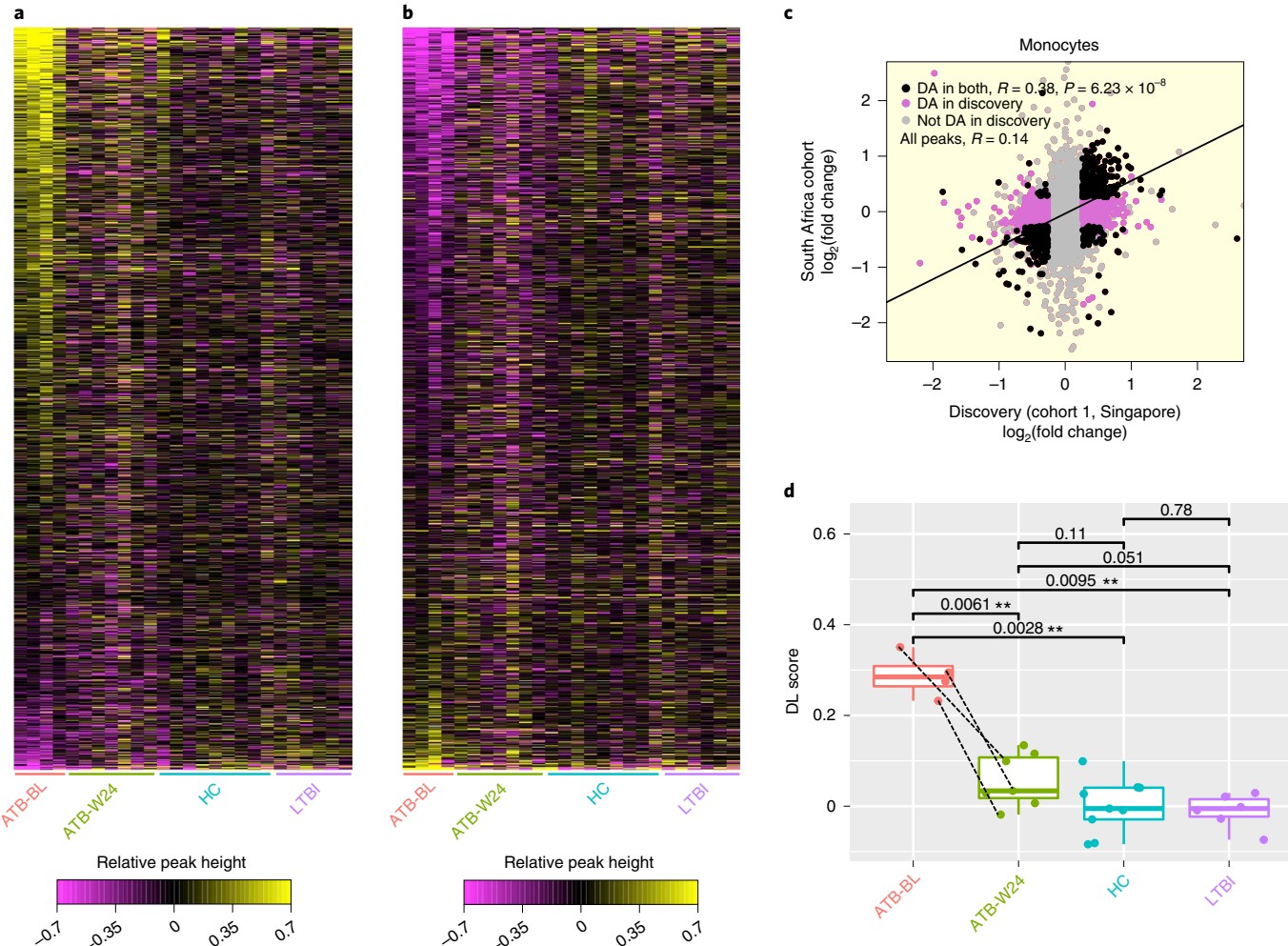

**Fig. 3 | Reproducibility of differential H3K27 acetylation in monocytes. a**, Each row represents an H3K27ac peak that showed increased acetylation in ATB relative to HC in the Singapore discovery cohort ('Up' DA peaks). Each column represents a sample in the longitudinal South African cohort (Cohort 4). The heat map is coloured on the basis of relative peak height, calculated as: $\log_2$(peak height in sample) – average($\log_2$(peak height in HC)). ATB-BL, ATB at baseline (untreated); ATB-W24, ATB after 24 weeks of treatment. **b**, Same as **a**, but for peaks with decreased acetylation in ATB relative to HC in the Singapore discovery cohort ('Down' DA peaks). **c**, Scatterplot of ATB-vs-HC peak height fold change in the Singapore discovery cohort and the South African cohort. **d**, The differential log(peak height) values shown in **b** were averaged across all Up DA peaks to calculate a differential log (DL) score for each sample. Dashed lines indicate samples from the same individual. Two-sided *t*-test, **$P < 0.01$. ATB-BL $n = 4$, ATB-W24 $n = 7$, HC $n = 9$, LTBI $n = 6$. The box shows the 25th and 75th percentiles, the median is indicated by a thick horizontal line, and the ends of the whiskers (segments) indicate 1.5 times the interquartile range.

corresponded to differentially expressed genes, those from the ETS and NF-κB families did not, suggesting that the observed chromatin alterations could be mediated by a combination of steady-state and differentially expressed TFs.

To identify potential feedback loops in the gene regulatory network formed by the above-mentioned TFs, we used hypergeometric optimization of motif enrichment (HOMER) to scan for occurrences of their DNA-binding motifs in their flanking DA peaks. Interestingly, we found that these TFs appeared to form multiple autoregulatory and cross-regulatory feedback loops in granulocytes (Extended Data Fig. 3a). Although we did not detect such autoregulatory loops in monocytes, it is possible that they exist even in this cell type. Next we inspected whether the DA peaks surrounding the *KCNJ15* gene contained the above-mentioned TF binding sites. Indeed, we found 5 such TF motifs, including IRF7, STAT1 and STAT5A to be present in the *KCNJ15*-associated enhancers, suggesting that they may regulate *KCNJ15* during the host response (Extended Data Fig. 3b).

Next we used GOrilla[22] to detect functional categories enriched in DE genes (fold change $\geq 1.5$, false discovery rate (FDR) *Q*-value $\leq 0.05$). Genes upregulated in granulocytes showed greatest enrichment for 'defence response to other organism' ($P = 2.2 \times 10^{-28}$; Fig. 4c), including genes associated with viral defence and innate immune response (Supplementary Tables 29 and 30). The latter two functional categories were also strongly enriched for DA peaks upregulated in granulocytes (Fig. 4a). These results are consistent with gene expression changes previously observed in whole blood from ATB patients, and also with the known role of type I interferon in host response to *Mtb*[4–6,10,23]. Upregulated genes in monocytes were enriched for 'regulation of multicellular organismal process' ($P = 3.4 \times 10^{-16}$; Fig. 4c), which included genes associated with inflammatory response and regulation of response to external stimulus (Supplementary Table 31).

To identify the most extensively altered chromatin loci in TB, we tested whether the peaks near individual genes were enriched for differential acetylation (Supplementary Table 32). *FNDC1* and

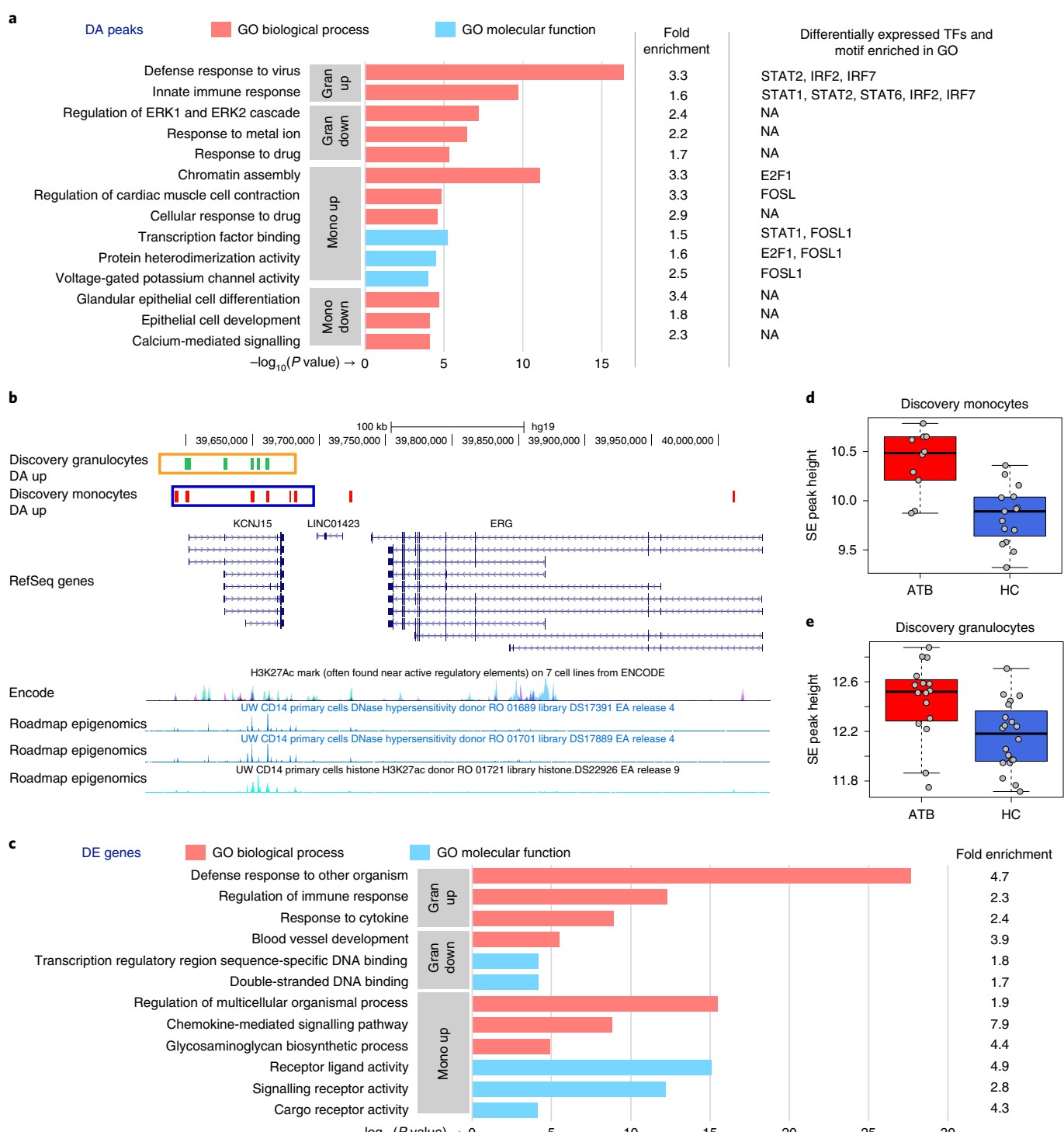

**Fig. 4 | Functional-enrichment analysis of DA peaks, KCNJ15 UCSC browser view and SE peak heights. a,** Discovery cohort. Gene ontology enrichment analysis of granulocyte and monocyte DA peaks relative to all peaks. The top three non-redundant gene ontology terms (by *P* value) are shown. For each term, the fold enrichment of DA peaks is shown, as well as enriched TF motifs and corresponding differentially expressed TFs (if any). N/A, not applicable. **b,** *KCNJ15* locus. Green and red ticks indicate DA peaks upregulated in response to infection in granulocytes and monocytes, respectively. Orange and blue boxes indicate the corresponding super-enhancer (SE) regions. For reference, H3K27ac ChIP-seq and DNaseI hypersensitivity data from CD14 primary cells (Roadmap Epigenomics) are shown below, along with layered H3K27ac from 7 cell types (ENCODE). **c,** Gene ontology enrichment analysis of DE genes, similar to **a**. **d,** SE peak heights of monocyte discovery samples at the *KCNJ15* locus. The peak heights were calculated from reads that mapped into the monocyte SE region displayed in **b**, and covariates were regressed using the same method used for the RNA-seq data (Methods). The boxplot shows that the SE is activated in response to TB infection (unpaired two-sided Wilcoxon test $P = 8.99 \times 10^{-4}$; fold change, 1.5; ATB $n = 10$, HC $n = 15$). The box shows the 25th and 75th percentiles, the median is indicated by a thick horizontal line, and the ends of the whiskers (segments) indicate 1.5 times the interquartile range. **e,** Similar to **d**, granulocyte samples (Wilcoxon test $P = 3.48 \times 10^{-3}$; fold change, 1.3; ATB $n = 16$, HC $n = 20$).

*TAGAP*, two neighbouring genes involved in G-protein signalling, showed the greatest enrichment for peaks upregulated in granulocytes. Thus, this locus, which is known for association with non-infectious inflammatory diseases such as Crohn's disease and coeliac disease[24], may also participate in the inflammatory response of granulocytes during *Mtb* infection. Among genes with greatest enrichment for downregulated peaks in granulocytes, *SGK1* was the most significantly altered. This gene plays a role in granulocyte apoptosis, and its downregulation leads to a prolonged state of inflammation[25]. *AGPAT4*, the gene most significantly enriched in monocyte downregulated peaks, transfers acyl chains in diacylglycerol biosynthesis, thus making them available for further use. Intriguingly, *Mtb* utilizes host acyl chains when it acquires a non-replicating-like state in macrophages[26], a condition important for *Mtb* survival. The gene most significantly associated with monocyte upregulated peaks was the above-mentioned potassium channel gene *KCNJ15*.

We then used HOMER[27] to scan for super-enhancers (SEs)[28] in ChIP-seq profiles from the discovery cohort. SEs are known to strongly activate gene expression in a cell-type-specific manner. They are thus probably important for the functioning of the corresponding cell type, although it is not clear if they are functionally distinct from regular enhancers that drive strong gene expression[29]. We identified 757 SEs in granulocytes and 905 in monocytes (Supplementary Tables 33 and 34). Notably, many of the DA peak clusters overlapped SEs (Supplementary Table 32), which further supports the significance of the corresponding DA loci in host response to ATB. We note that these overlaps are not indicative of a pattern of enrichment (Supplementary Table 35). In particular, the upregulated monocyte DA peak cluster near *KCNJ15* was mostly contained within an SE that also showed significantly increased acetylation in ATB (Fig. 4b,d,e).

**Role of *KCNJ15* in host response to TB.** We prioritized *KCNJ15* for functional validation because (1) this locus showed the greatest enrichment for upregulated monocyte DA peaks, (2) the DA peaks coincided with an SE, (3) its role in monocyte physiology, infection and inflammation has not been previously studied and (4) potassium channels have not been studied in the context of mycobacterial infection. *KCNJ15* encodes the poorly characterized two-transmembrane tetrameric inward-rectifier-type potassium (K[+]) channel Kir4.2[30–32], which potentially allows K[+] to flow into cells when hyperpolarized[33]. To corroborate the increased histone acetylation at *KCNJ15* in granulocytes and monocytes (Fig. 4b,d,e and Supplementary Table 32), we analysed *KCNJ15* mRNA expression in five different TB cohorts (Supplementary Table 36).

In all five cohorts, *KCNJ15* mRNA levels in granulocytes, monocytes or whole blood were significantly higher in ATB than in HC, with monocytes showing the greatest change (Fig. 5a). In the UK dataset, *KCNJ15* expression in whole blood was attenuated after 2 months of treatment and reached levels comparable to those of healthy individuals within 12 months (Fig. 5a). *KCNJ15* expression was also found to be significantly upregulated in whole blood from Thai ATB patients[34]. Despite the consistent evidence from transcriptomics studies, *KCNJ15* has not previously been prioritized for mechanistic analysis of host response to TB, perhaps because other genes showed even greater differential expression. In terms of histone acetylation response, however, this locus is a clear outlier and thus a strong candidate for functional studies.

To investigate the causal relationship between mycobacterial infection and *KCNJ15* expression, we infected primary monocytes and THP-1 cells with *Mtb* and BCG (Bacillus Calmette–Guérin), respectively, and observed increased mRNA and protein expression upon infection (Extended Data Fig. 4a–c). Interestingly, in THP-1 cells cultured in standard medium, Kir4.2 was localized to the cell membrane (Extended Data Fig. 4c) and BCG-containing lysosomes (Fig. 5b,c and Supplementary Video 1). Lysosomal localization of Kir4.2 may be attributable to its three lysosomal-targeting dileucine motifs ([44]DGIYLL[49], [244]ESPFLI[249] and [364]ELRTLL[369]). In uninfected cells, Kir4.2 showed low expression and low APG-4 fluorescence (indicator of intracellular potassium concentration [K[+]]$_i$; Fig. 5c). In contrast, BCG-infected cells showed higher Kir4.2 expression and intense and concentrated APG-4 fluorescence in phagosomes (Fig. 5c). Kir4.2 was localized to BCG-containing phagosomes, which could include late endosomes as well as lysosomes (that is, endo-lysosomes). The proportion of BCG residing in acidic vacuoles (endo-lysosomes) was ~50%, with the remaining fraction presumably residing in earlier endosomes that were not stained by LTR (Lysotracker; Extended Data Fig. 4d–f). Overall, these results suggest that mycobacterial infection is associated with Kir4.2 upregulation and its localization to the phagosome, resulting in increased K[+] in phagosomes.

To assess the impact of *KCNJ15*/Kir4.2 on *Mtb* growth, we performed gain and loss of function experiments. Patch-clamp analysis of the THP-1 cell line stably overexpressing *KCNJ15* (KCNJ15[OE]) revealed inwardly rectifying K[+] currents that could be blocked by Ba[2+] in a voltage-dependent fashion (Fig. 5d and Supplementary Fig. 7). These KCNJ15[OE] THP-1 cells restricted mycobacterial growth (Fig. 5e). Consistently, THP-1 cells in which *KCNJ15* was knocked down (KCNJ15[KD]; Extended Data Fig. 4g,h) were less efficient at controlling *Mtb* growth (Fig. 5f). Furthermore, lentivirally-mediated overexpression of Kir4.2 in primary monocytes (KCNJ15[++]; Extended

**Fig. 5 | Functional characterization of *KCNJ15*/Kir4.2. a**, *KCNJ15* mRNA expression in ATB and HC individuals from multiple cohorts (Supplementary Table 36). Singapore: RNA-seq, this study; UK 2010, Gambia 2011: microarray. Red line, median. *P* value: two-tailed Mann–Whitney *U* test (case-control) and paired Wilcoxon signed-rank test (time course). **b**, 3D projection of a THP-1 monocyte infected with mcherry-BCG (red), stained for lysosomes (cyan) and *KCNJ15*/Kir4.2 (green). The images were acquired 24 h post infection by 3D-SIM. Scale bar, 2 μm. **c**, Confocal images of uninfected and mcherry-BCG-infected (blue pseudocolor) THP-1 monocytes stained for intracellular K[+] (APG-4, green) and Kir4.2 (red), 24 h post infection. Scale bar, 2 μm. Data from 3 independent experiments. **d**, Left: patch-clamp recordings of THP-1 Control[OE] and KCNJ15[OE] cells using a ramp protocol from −120 mV to 120 mV in 200 ms showing currents elicited at 4.5, 50 and 160 mM [K[+]]$_e$. Right: average current amplitude recorded at −100 mV in 50 mM [K[+]]$_e$ is shown as a scatterplot, *n* = 7 and 13 cells for Control[OE] and KCNJ15[OE], respectively. Data from 2 independent experiments. **e**, Mycobacterial (BCG) growth in THP-1 monocytes; compiled data of *n* = 3 independent experiments, each in triplicate. **f**, *Mtb* growth after 24 h in scrambled control (−) and KCNJ15[KD] (+) THP-1 monocytes; compiled data of *n* = 3 independent experiments, each in triplicate. **g**, *Mtb* growth in *KCNJ15*-overexpressing (KCNJ15[++]) and control (empty vector) primary monocytes; *n* = 4 donors, 24 h post infection. CFU, colony-forming units. *n* = 2 independent experiments. **h**, Volcano plot of differential gene expression between KCNJ15[++] and control cells. Apoptotic genes have been highlighted. *P* values from two-sided Wald test in DESeq2 package, FDR < 0.05. **i–k**, Flow cytometry data of Control[OE] and KCNJ15[OE] THP-1 monocytes stained with Annexin V and PI. Data from 2 independent experiments, *n* = 6. **l**, Immunoblot of protein lysate of Control[OE] and KCNJ15[OE] THP-1 cells with APAF1 and β-actin. Data from 2 independent experiments. **m**, Control[OE] and KCNJ15[OE] THP-1 cells stained for mROS using Mitosox red. Mean fluorescence intensity (MFI) data from *n* = 150 cells of either type. Top panel-representative images. Bottom panel-compiled data. *P* values in **d**, **e** and **i–k** were calculated using two-sided unpaired *t*-test, two-sided paired *t*-test in **f**, and two-sided Mann–Whitney *U* test in **g** and **m**. In **d–f**, mean ± s.e.m. Data in **g** and **i–k** are shown as box and whiskers, minimum to maximum. *\*P* <= 0.05, *\*\*P* <= 0.01, *\*\*\*P* <= 0.001, *\*\*\*\*P* <= 0.0001; NS, not statistically significant.

Data Fig. 4i,j) also showed *Mtb* growth inhibition (Fig. 5g). These results suggest that overexpression of *KCNJ15* upon *Mtb* infection is a protective host response.

To identify potential mechanisms underlying the inhibition of mycobacterial growth by cells overexpressing Kir4.2, we performed RNA-sequencing on KCNJ15++ and control primary monocytes.

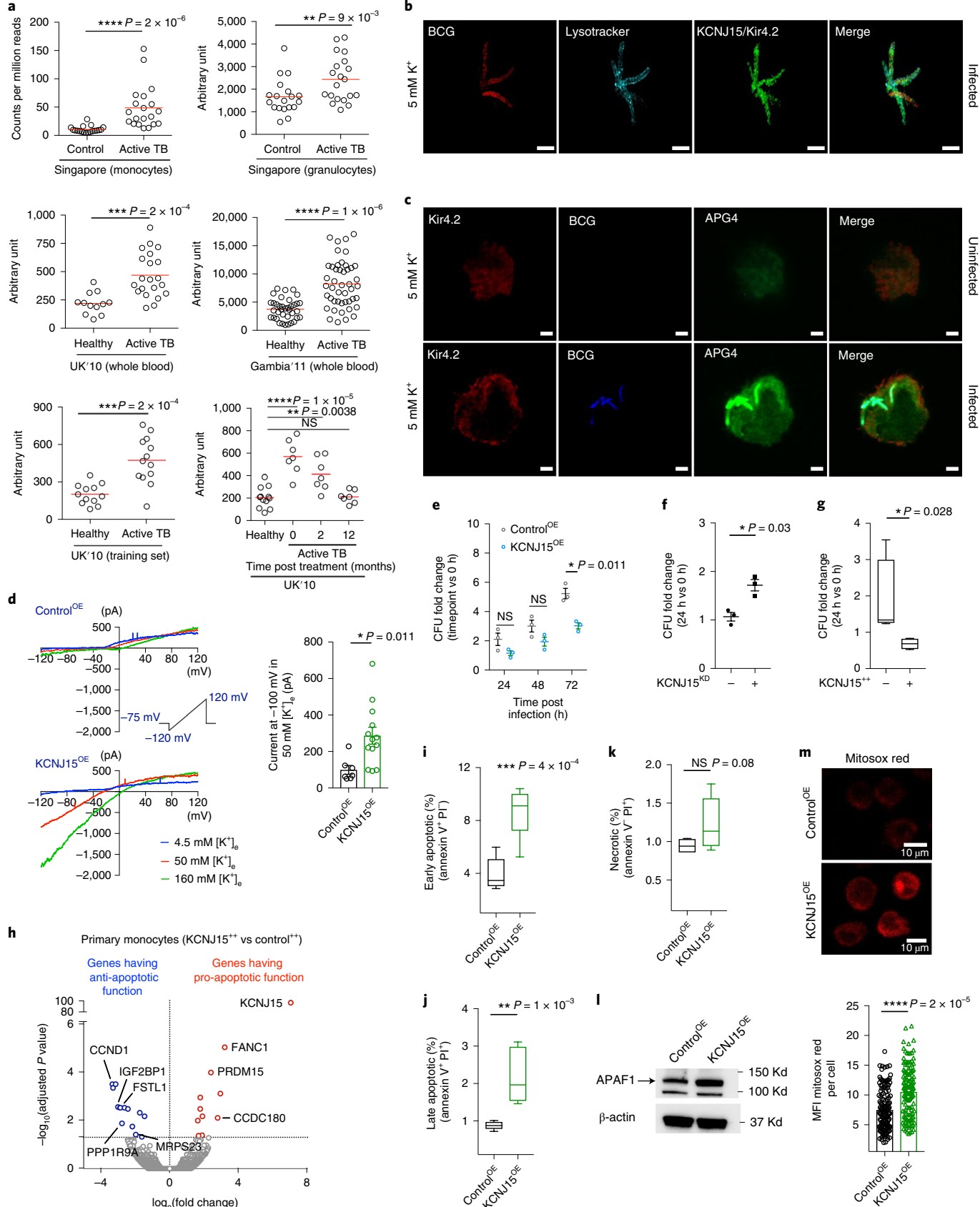

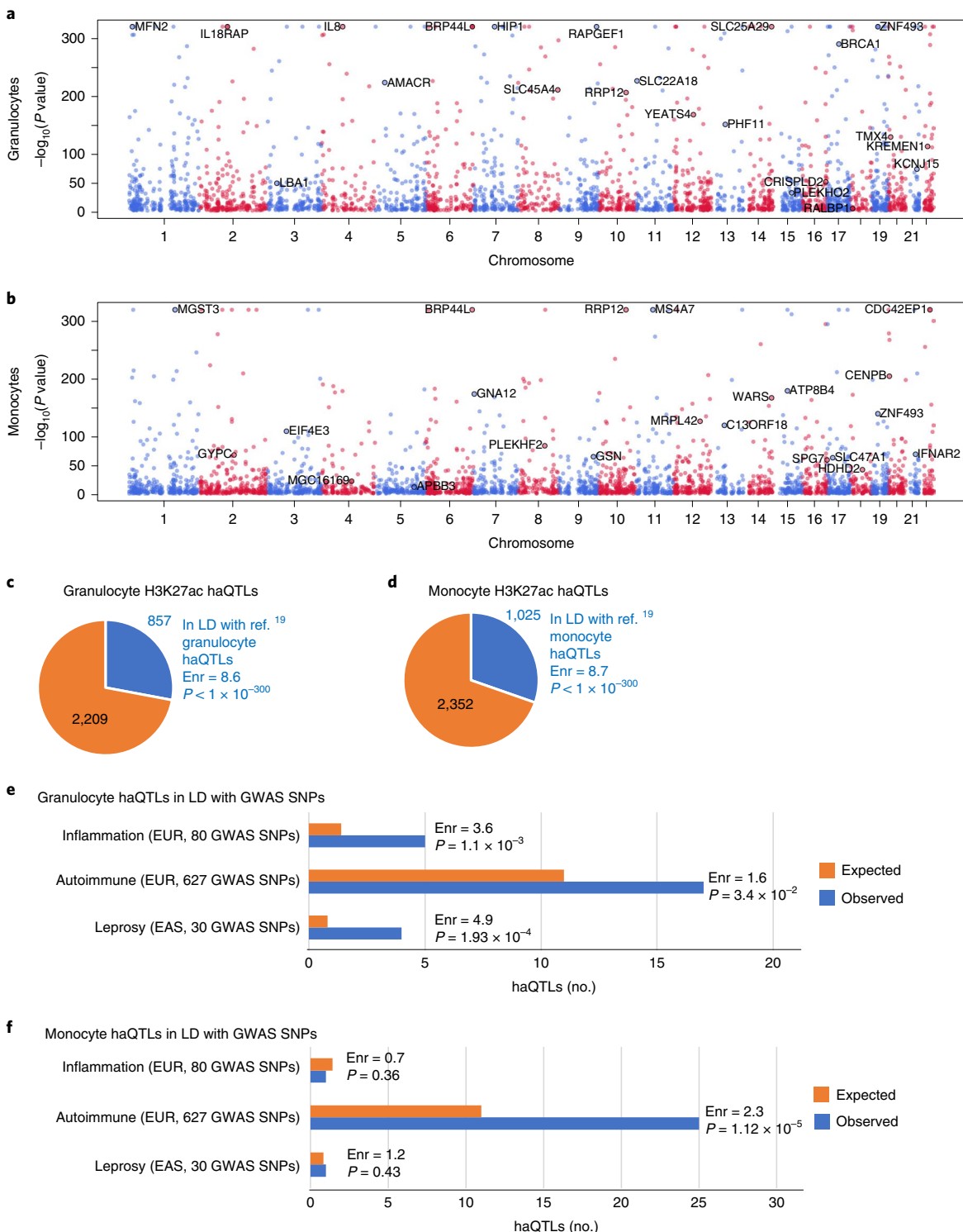

**Fig. 6 | Landscape of haQTLs in granulocytes and monocytes. a,b,** Manhattan plots of haQTL distribution in granulocytes and monocytes. Gene names are indicated next to haQTLs that are in LD with eQTLs in the same cell type. Granulocyte eQTLs were obtained from Naranbhai et al.[45] and monocyte eQTLs from Fairfax et al.[46] The alternating red and blue colours of the dots are used to delineate chromosomes. *P* values are from the G-SCI test (details in Supplementary Methods, 'Histone Acetylation QTLs'). **c,d,** Pie charts of granulocyte and monocyte haQTLs identified in this study that are in LD with haQTLs from a previous study. Fold enrichment (Enr) and *P* value of haQTL overlap relative to randomly chosen SNPs in peaks are indicated (details in Supplementary Methods, 'Overlap with published haQTL datasets'). **e,f,** Number of non-redundant granulocyte and monocyte haQTLs in LD with genome-wide significant GWAS SNPs. Orange bars, expected number of haQTLs adjusted for minor allelic frequency distribution and distance to the nearest transcription start site; blue bars, observed; *P* values, *Z*-score test (details in Methods, 'Statistical significance of LD between haQTLs and GWAS SNPs'); EUR, European; EAS, East Asian.

A total of 25 genes were found to be differentially expressed in KCNJ15++ cells (Supplementary Table 37). Multiple downregulated genes such as *CCND1*, *IGF2BP1*, *FSTL1*, *PPP1R9A*, and *MRPS23* (Fig. 5h), have been shown to be anti-apoptotic in nature (Supplementary Table 37), and upregulated genes such as *CCDC180*, *FANC1*, and *PRDM15* (Fig. 5h), have possible pro-apoptotic properties (Supplementary Table 37). This suggests that cells overexpressing *KCNJ15* may show increased apoptosis. Indeed KCNJ15^OE cells were enriched for early and late apoptosis (Fig. 5i,j and Supplementary Fig. 8). However, there was no indication of necrosis or altered growth in KCNJ15^OE cells (Fig. 5k and Extended Data Fig. 4k). Apoptosis has been shown to play an important role in host defence against *Mtb*[35]. Since mitochondria play a key role in activating apoptosis in mammalian cells, we quantified (i) the expression of APAF1 (apoptotic protease activating factor-1), a crucial pro-apoptotic molecule in the mitochondrial apoptosis pathway[36], and (ii) mitochondrial ROS (mROS), a factor that potentiates apoptosis[37] and restricts growth of bacteria, including *Mtb*[38,39]. KCNJ15^OE cells were found to have increased APAF1 protein levels (Fig. 5l) and also increased mROS (Fig. 5m). Overall, our results indicate that Kir4.2 is a conduit in the cell membrane for K+ to enter monocytes, and its upregulation contributes to a protective host response to *Mtb* infection.

**Genetic analysis of haQTLs and GWAS.** In addition to identifying disease-associated chromatin changes, HAWAS data can be used in conjunction with the genotype-independent signal correlation and imbalance (G-SCI) test[40] to sensitively detect SNPs associated with population variation in histone acetylation, that is haQTLs[14]. These haQTLs could then be used to identify candidate causal variants in GWAS loci[14,19,40–44]. We identified 3,066 haQTLs in granulocytes and 3,377 in monocytes (FDR *Q*-value≤0.05; Fig. 6a,b and Supplementary Tables 38 and 39). Note that disease status was regressed out before QTL analysis (Methods), and thus the resulting haQTLs are not specific to TB. The haQTLs from our discovery cohort (exclusively Chinese ethnicity) were strongly enriched for high linkage disequilibrium (LD) with haQTLs previously detected in granulocytes and monocytes from a European cohort[19]. Comparison of effect sizes (that is the regression beta) of haQTLs between the Singapore discovery and European cohorts revealed a strong positive correlation (granulocytes: R = 0.68, $P < 2.2 \times 10^{-16}$; monocytes: $R = 0.73$, $P < 2.2 \times 10^{-16}$; Extended Data Fig. 5a,b), indicating that the direction of effect was highly concordant.

Despite the high concordance between genetically linked haQTLs, >75% of our haQTLs were novel (Fig. 6c,d). We examined allele frequency differences between shared and non-shared haQTLs, as this could potentially explain the large fraction of novel QTLs discovered. In general, it is expected that haQTLs with higher minor-allele frequency would more likely be detected in any given cohort, due to higher statistical power. Consequently, we hypothesized that high-frequency haQTLs would be enriched in the shared subset, and this was indeed the case (Extended Data Fig. 5c). Thus, the lower detectability of haQTLs with low minor-allele frequency in either cohort may have contributed to the lack of complete overlap between the two haQTL sets. In addition, numerous other factors may also have contributed, including differences in ethnicity, age and environment, differences in experimental protocols, differences in sample quality and differences in bioinformatic pipelines. We also note that our haQTL analysis approach exploited additional information present in the data, namely allelic imbalance in ChIP-seq reads at heterozygous sites. By modelling the effect of genotype on both peak height and allelic imbalance, our approach had relatively high statistical power. We have previously shown that incorporation of allelic imbalance in the test can increase the number of detected haQTLs by up to 3-fold (G-SCI test)[40]. Furthermore, our haQTLs co-localized with eQTLs previously detected in granulocytes and monocytes[19,45,46], indicating they may contribute to gene expression variation (Extended Data Fig. 6a–d and Supplementary Tables 40–45). Finally, we compared the effect sizes of our haQTLs to those of linked eQTLs in the European IHEC dataset[19] (Extended Data Fig. 6e,f). We found that haQTL and eQTL effect sizes were strongly correlated, further supporting the robustness of our haQTL set.

To identify haQTLs that may contribute to human phenotypes, we tested them for LD with non-coding GWAS variants ($P \leq 5 \times 10^{-8}$) for infectious diseases, inflammation and autoimmune disorders[47] (Methods; Supplementary Tables 46–52). None of the 23 TB-associated SNPs from the GWAS catalogue were in LD with our haQTLs, perhaps because 20/23 were detected in non-Asian populations[48] (Supplementary Table 46). We then examined the 30 Asian GWAS SNPs associated with leprosy, which is caused by *Mycobacterium leprae*. These 30 SNPs were enriched for LD with granulocyte haQTLs ($P = 1.9 \times 10^{-4}$; Fig. 6e and Supplementary Tables 47 and 48), suggesting that some of the granulocyte haQTLs may contribute to mycobacterial infection by altering gene regulation in host cells[4]. We also tested the haQTL sets for association with GWAS SNPs for inflammation and autoimmune disorders (Supplementary Tables 49–52). Most prominently, monocyte haQTLs were enriched for LD with autoimmune GWAS SNPs ($P = 1.1 \times 10^{-5}$; Fig. 6f). Granulocyte haQTLs showed a similar, though lower, enrichment. In addition, granulocyte haQTLs were significantly enriched in GWAS loci related to inflammation ($P = 1.1 \times 10^{-3}$). Thus, the haQTL sets probably contain multiple genetic variants contributing to inflammation and autoimmune disorders.

We next examined the contribution of all SNPs in all peaks, or DA peaks, to TB heritability. We found that one of the 23 TB-associated SNPs from the GWAS catalogue (rs9272785) was in LD with 16 SNPs residing within 5 monocyte acetylation peaks, one of which is a DA peak (Supplementary Table 53). In granulocyte peaks, we found that the same TB-associated SNP (rs9272785) was in LD with 6 SNPs that lie within a granulocyte peak that was not DA (Supplementary Table 53). Next, we estimated the proportion of heritability residing within acetylation peaks using the largest TB GWAS whose summary statistics were available in the NHGRI GWAS Catalogue[49]. Using stratified LD-score regression[50], we estimated that granulocyte and monocyte peaks explained 12.6% and 33.2% of TB heritability, respectively (Supplementary Methods). In contrast, DA peaks did not explain TB heritability (<1% for both granulocyte and monocyte DA peaks), most probably due to the small fraction of the genome they covered[16] (~0.1%). In summary, genetic variation in cis-regulatory sequences of granulocytes and monocytes explained a substantial proportion of the heritability of TB.

## Discussion

Our study broadens the scope of the HAWAS approach by demonstrating its applicability to an immune-related phenotype. Since previous HAWAS analyses were based on a single cohort, the reproducibility of the detected DA peaks could not be established. Our comparison of Singapore discovery and validation cohorts demonstrates that DA peaks are indeed reproducible, despite potential confounders such as age, sex, ethnic diversity and technical variation. We further demonstrated that the majority (86%) of monocyte DA peaks from the Singapore discovery cohort were reproducible in an independent South African cohort despite ethnic, geographical and technical differences between the two datasets (Fig. 3). It is probable that the SA patients were potentially infected with distinct *Mtb* strains relative to the Singapore patients (although molecular typing data were unavailable), indicating that chromatin changes may be mostly independent of strain. Importantly, this result was obtained despite a relatively small cohort size, of the order of 20 disease vs 20 control samples. Thus, HAWAS is an inexpensive and

robust technique, which could potentially offer mechanistic insights into diverse diseases[14,16–18].

Data from the South African cohort allowed us to examine two additional dimensions of host chromatin response, namely drug response and distinctions between active and latent TB. After ATB patients received 24 weeks of drug treatment, histone acetylation at DA peaks reverted to levels close to those seen in HC, suggesting that the host chromatin response largely subsides upon clearance of the pathogen (Fig. 3b–d). No global difference in chromatin state was observed between LTBI and HC, suggesting that the chromatin alterations at DA peaks arise mostly during active *Mtb* infection.

In addition to identifying disease-associated chromatin traits (DA peaks), HAWAS provides a rich resource of haQTLs, some of which may represent the underlying causal variants in GWAS loci. Overall, only ~30% of our Asian granulocyte and monocyte haQTLs were in LD with corresponding European haQTLs (Fig. 6c,d), suggesting that haQTL analysis in novel populations could substantially expand the set of known regulatory variants and thus aid in the prioritization of causal candidates in GWAS loci. For example, Asian-specific haQTLs provided novel candidate variants in gene loci associated with leprosy (*SIGLEC5*, *TNFSF15*) and autoimmune disease (*CARD9*, *IRF5*, *IL27*, *FOS*) (Supplementary Tables 47 and 48). Although the small set of known TB GWAS loci showed limited overlap with our haQTLs, H3K27ac peaks as a whole explained a substantial proportion of TB heritability (granulocytes 12.6%, monocytes 33.2%). In summary, cohort-scale chromatin profiling in novel populations provides multiple insights into disease genetics.

The >2,000 differentially acetylated loci identified in this HAWAS provide a large set of candidates for locus-specific mechanistic investigations into *Mtb*–induced host response. The enrichment of DA peaks near voltage-gated potassium channel genes of the KCNJ family is particularly intriguing. Although this gene category has not previously been examined in the context of innate immune response to mycobacterial infection, the connection to TB is plausible because potassium channels are thought to stimulate apoptosis[51], lysosomal function[52] and nitric oxide production[53], all of which are known mechanisms of *Mtb* control by monocytes and macrophages[35,54]. Furthermore, these anti-microbial functions of potassium channels could be due to the regulation of $[K^+]_i$, which has been implicated in host–pathogen interaction[55]. Indeed, we found that (1) knockdown of *KCNJ15*/Kir4.2 increased mycobacterial growth in monocytes and (2) upregulation of Kir4.2 decreased mycobacterial growth by inducing an apoptotic (viz. protective) host response. Since potassium channels are druggable, these results open up a new avenue for research into host-directed therapies for TB[56].

## Methods

**Human participants.** *Singapore participants (discovery cohort/Cohort 1, validation cohort/Cohort 2, expression cohort/Cohort 3).* HIV-negative ATB patients (based on clinical diagnosis with mycobacterial and radiographic evidence) and HC individuals (negative for IFN-γ release assay (IGRA), QuantiFERON TB gold test) were recruited at Tan Tock Seng Hospital's Tuberculosis Control Unit (TTSH, TBCU). Additional HC individuals were recruited locally at Singapore Immunology Network (SIgN). Patients were sampled within 4 d of anti-TB treatment initiation and excluded if they had previously received anti-TB therapy. This study was approved by the Domain Specific Review Board of the National Healthcare Group (2010/00566) and Institutional Review Board of the National University of Singapore (09-256). All participants provided written informed consent. To create the discovery cohort, we chose age-matched participants with Chinese ethnicity and the remaining samples were denoted as the validation cohort (for details see 'Read alignment, peak calling and peak height normalization' below). Cohort 3 constituted an independent recruitment for RNA-sequencing and all samples that passed RNA-seq quality control were included in the analysis (for details see 'RNA-sequencing (Cohort 3)' below).

*South African cohort (longitudinal cohort/Cohort 4).* Participants were recruited from the Ubuntu Clinic, Site B, Khayelitsha (Cape Town, South Africa) between March 2017 and December 2018. All participants were HIV-uninfected adults (age ≥ 18 yr) and provided written informed consent. The study was approved

by the University of Cape Town Human Research Ethics Committee (HREC, 050/2015) and was conducted under DMID protocol no.15-0047. Participants were grouped according to their TB activity status:

1. ATB: all ATB patients (median age 27.3, interquartile range (IQR) 23–33, 60% male) tested sputum Xpert MTB/RIF (Xpert, Cepheid)-positive and had clinical symptoms and/or radiographic evidence of TB. All ATB individuals were drug sensitive and had received no more than one dose of anti-tubercular treatment at the time of baseline blood sampling. A follow-up sample was obtained at completion of TB treatment (24 weeks).

2. LTBI: all individuals with latent *Mtb* infection (median age 26.5, IQR 23–33, 60% male) were asymptomatic, had a positive IGRA (QuantiFERON TB Gold In-Tube, Qiagen), tested sputum Xpert MTB/RIF-negative and exhibited no clinical evidence of active TB.

3. HC: healthy controls (median age 30.5, IQR 24–32, 40% male) were negative for IGRA, tested sputum Xpert MTB/RIF-negative and exhibited no clinical evidence of ATB.

In all cohorts, no compensation was given to any participant.

*Reagents.* The following chemicals were used: KCl (Sigma-Aldrich, P954), paraformaldehyde (Electron Microscopy Sciences, 15710). The following antibodies were used: anti-KCNJ15 (Sigma-Aldrich, HPA016702), anti-APAF-1 (Cell Signaling Technology, 5088), anti–glyceraldehyde-3-phosphate dehydrogenase (GAPDH) (14C10) (Cell Signaling Technology, 2118), anti-β-actin (Cell Signaling Technology, 4967), anti-rabbit immunoglobulin G (IgG) horseradish peroxidase (HRP)-linked antibody (Cell Signaling Technology, 7074). ON-TARGET plus SMARTpool Human KCNJ15 (3772) small interfering RNAs (siRNAs) (L006245000005) and control siRNAs (1299001) were from Dharmacon and Integrated DNA Technologies, respectively.

*THP-1 cell line.* Human monocyte THP-1 cells from American Type Culture Collection were maintained in Roswell Park Memorial Institute (RPMI)-1640 medium (Gibco), supplemented with 10% heat-inactivated foetal bovine serum (FBS), 1% sodium pyruvate, 1% L-glutamine, 1% non-essential amino acids (Life Technologies) and 1% kanamycin (Sigma-Aldrich) at 37 °C in a 5% $CO_2$ humidified atmosphere. In infection experiments, no antibiotic was used. Macrophage differentiation of THP-1 cells was induced by treating with 100 ng ml⁻¹ phorbol myristate acetate (PMA) for 24 h. Cells were then washed and rested for 24 h before infection.

*Primary monocytes and granulocytes.* Primary cells were purified from the blood of participants recruited in this study. PBMCs and granulocytes were separated using a Ficoll gradient and red blood cells were lysed were lysed. From PBMCs, monocytes were isolated using CD14⁺ immunomagnetic separation beads (MACS, Miltenyi). Isolated cells were immediately fixed with 1.6% paraformaldehyde and stored at −80 °C before processing for Chip-seq. Flow cytometric analysis was performed on purified cell populations using monoclonal antibodies against CD3, CD14 and CD15. Both cell populations showed purity >95% (Supplementary Fig. 9).

*ChIP-seq of isolated granulocytes and monocytes.* For each ChIP-seq experiment on Singapore samples, approximately 10 million fixed granulocytes or 3 million monocytes were thawed on ice. The starting amount for monocyte samples from South Africa was 0.5 million. Cells were lysed (10 mM Tris-HCl (pH 8), 0.25% Triton X-100, 10 mM EDTA, 100 mM NaCl, Roche 1X Complete Protease Inhibitor) and nuclei were collected and resuspended in 300 µl SDS lysis buffer (1% SDS, 1% Triton X-100, 2 mM EDTA, 50 mM Hepes-KOH (pH 7.5), 0.1% sodium deoxycholate, Roche 1X Complete Protease Inhibitor). Nuclei were lysed for 15 min, followed by sonication to fragment chromatin to an average size of 200–500 bp (Bioruptor NGS, Diagenode). Protein–DNA complexes were immuno-precipitated using 3 µg of H3K27ac antibody from the same lot for all ChIP experiments (Active Motif, 39133) coupled to 50 µl Protein G Dynabeads (ThermoFisher) overnight. Beads were washed and protein–DNA complexes were eluted using 150 µl of elution buffer (1% SDS, 10 mM EDTA, 50 mM Tris-HCl (pH 8)), followed by protease treatment and de-crosslinking overnight at 65 °C. After phenol/chloroform extraction, DNA was purified by ethanol precipitation. Library preparation was performed as described previously[14]. After 15 cycles of PCR using indexing primers, libraries were size selected for 300–500 bp on low-melting agarose gel and 4 libraries were pooled for sequencing at 2×100 bp in each lane of an Illumina HiSeq 2000 flow cell using Version 3 reagents. In total, we generated 230 H3K27ac ChIP-seq datasets from 90 granulocyte and 140 monocyte samples (100 Singaporean discovery and validation, 40 from the longitudinal South African cohort).

*Read alignment, peak calling and peak height normalization.* Reads were mapped to the human genome (GRCh37) using BWA-0.7.5a[57] at default parameter settings as described previously[14,40]. Reads not annotated by BWA as properly paired were discarded, as were reads with mapping quality <10. Duplicate reads (read-pairs mapping to the same genomic location) were collapsed. For each sample, ChIP-seq peaks were detected relative to input control using DFilter[13] at a *P* value threshold of 1×10⁻⁶. For either cell type, the initial peak set was defined as the union of

peaks from the entire set of discovery and validation samples from Batches 1 and 2, which included the vast majority of samples (Supplementary Table 1). Peaks wider than 8 Kb were then discarded. We performed multi-sample correction for GC bias[13,40] separately on each processing batch. We then discarded samples with low complexity as measured by the number of unique read-pairs (<15 million for Batches 1 and 2; <30 million for Batch 3). We also discarded samples with high GC bias (>4,500 peaks showing greater than two-fold GC bias in either direction).

Cohorts 1 and 2 (ChIP-seq, Singapore discovery and validation cohorts). Of the 135 retained high-quality samples (74 granulocyte, 61 monocyte), we designated those from age-matched Chinese donors in Batches 1 and 2 as belonging to the discovery cohort (Supplementary Table 1). The remaining samples, which were multi-ethnic (Chinese, Malay, Indian) and included a third processing batch, constituted the validation cohort. For each combination of cell type and cohort, the final consensus peak set was then defined as the set of peaks detected in at least 5 of the retained high-quality samples (Supplementary Tables 10–13). These consensus peaks presumably indicate the locations of active enhancers and promoters in the corresponding samples. Note that only autosomal chromosomes were considered in our analyses. Peak heights within each peak set were quantile normalized to match the distribution of mean peak height.

Cohort 4 (ChIP-seq, longitudinal South African cohort). H3K27ac ChIP-seq was performed on 40 monocyte samples and sequencing reads were aligned to the human genome as above. We discarded samples with less than 15 million unique read-pairs, samples with high GC bias (>4,500 peaks showing greater than two-fold GC bias in either direction), and samples whose peak regions total less than 25 Mb. After QC, 26 samples were retained for further analysis.

*Differential peak calling.* The following 7 biological and technical confounders were considered in our analysis of peak height variation across samples in the discovery cohort: age, sex, number of reads, median insert size, number of peaks, percent unique reads, and sequencing batch. We used an iterative strategy where the confounder that explained the greatest amount of variance was regressed out at each iteration (Extended Data Fig. 1a). Note that confounding factors were regressed out from the logarithm of the peak height matrix, following the multiplicative effects model used previously[14]. The procedure was terminated when the first 5 principal components of the residual peak height matrix were uncorrelated with any of the 7 potential confounders (Spearman correlation coefficient ≤0.4, Supplementary Fig. 2). This regression strategy accounts for the non-independence of confounding variables. By this procedure, we regressed out 4 covariates from the monocyte dataset (batch, no. of peaks, percent unique reads and sex) and 4 from the granulocyte dataset (batch, no. of reads, no. of peaks and sex). Data from the multi-ethnic validation cohort were controlled for two additional covariates: age and ethnicity (Supplementary Fig. 2). For each cohort, differentially acetylated peaks were determined using a 2-step approach[14]. First, using all individuals in the cohort, a differential acetylation $P$ value was calculated for each peak using the 2-sample two-sided Wilcoxon test (FDR $Q$-value ≤ 0.10, fold change ≥1.2). We then followed a procedure designed to restrict the DA peak analysis to the core set of consistent samples, that is, samples that displayed a global histone acetylation signature consistent with their categorization as ATB or HC[14]. Briefly, we defined the pairwise distance between two samples to be the Pearson correlation coefficient of their peak height vectors (DA peaks only). For each sample, we then calculated its median distance from HC samples and its median distance from ATB samples within the same cohort. If an HC sample was closer to ATB samples, it was discarded from the core set. Similarly, ATB samples that were closer to HC were also discarded from the core set. The final core set used for DA peak calling comprised 47 ATB and 71 HC samples (Supplementary Table 1). The final set of DA peaks was then defined on the basis of samples from the core set, using the same statistical criteria as above (Supplementary Tables 3, 5, 7 and 9). Discovery-cohort DA peaks defined in this manner were independently corroborated using the validation cohort (Fig. 2b). Peak height heatmaps and Bland-Altman (MA) plots for samples from the core set are shown in Extended Data Figs. 1b and 2a.

*Statistical power analysis.* We randomly resampled peak heights (after log transformation and zero centring) from the data matrix of the Singapore discovery cohort to create simulated peaks. Peak heights designated as ATB were shifted up or down to simulate a specified fold change relative to HC. In this manner, we simulated a range of fold changes from 1.05 to 2.0. For each fold change, we simulated 10,000 differential peaks and then assessed the number of peaks $N$ that passed the $P$ value cut-off of the weakest DA peak in the actual data. Statistical power in detecting DA peaks was then estimated as $N/10,000$. The results of this simulation are shown as a function of cohort size, assuming an equal number of HC and ATB samples (Supplementary Fig. 6).

*Functional-enrichment analysis of differentially acetylated peaks.* We used GREAT[21] to separately determine the enrichment of gene categories in up- and downregulated peaks. Genes were associated with regulatory regions using the basal+extension association rule defined by GREAT. The statistical significance of

DA-peak enrichment near genes from any particular category (relative to all peaks) was calculated using Fisher's exact test (FDR $Q$-value ≤ 0.05; fold change ≥1.5). Enriched functional categories containing fewer than 6 genes flanking DA peaks were then discarded. Finally, for display in Fig. 4a, we discarded any enriched gene category if it was redundant with a higher-ranked (more significant) category. Redundancy between two functional categories was defined as overlap of ≥40% in the number of genes flanking DA peaks. The complete list of GREAT results, including redundant hits, is shown in Supplementary Tables 20–23. Enrichment analysis of DA peaks near individual genes (Supplementary Table 32) was performed using the same statistical method, except that in this case each peak was assigned to its two closest genes.

*Comparison with IHEC data.* We downloaded bigwig and peak files corresponding to 184 granulocyte and 173 monocyte H3K27ac ChIP-seq datasets from the IHEC Data Portal[58] (accession numbers of IHEC datasets are in Supplementary Table 54). We obtained the intersection of IHEC peaks with peaks in our data (26,429 granulocyte and 33,720 monocyte peaks common to both) and used the bigWigAveToBed tool to calculate peak heights in the IHEC dataset[59]. For comparison to IHEC, we converted BAM files from this study to bigwig format and then ran bigWigAveToBed on the same peak regions. For each peak region, we correlated mean peak height (across samples) in the IHEC dataset to mean peak height (across samples) in our dataset.

*Detection of SEs.* For each monocyte and granulocyte sample from the discovery cohort, SEs were ascertained using HOMER[27]. SE annotations for each cell type were merged by constructing the union set of SEs from individual samples. Elements of the union set that overlapped SEs in fewer than 5 individuals were discarded, resulting in a final set of 757 consensus SEs in granulocytes and 905 in monocytes. SE peak height in each sample was defined as the number of ChIP-seq reads mapped to the corresponding genomic region. The SE coordinates and read counts for each sample as calculated by HOMER are given in Supplementary Tables 33 and 34.

*Motif enrichment analysis in differentially acetylated peaks.* We performed motif enrichment analyses for each set of DA peaks (gran_up, gran_down, mono_up, mono_down) using HOMER and the transcription factor motif database (TRANSFAC)[60]. The background set used was all peaks and the resulting motifs were filtered with the following parameters (fold change ≥1.3, number of peaks with motifs >50, and FDR $Q$-value ≤0.05). Next we performed another motif enrichment analysis, this time using only peaks associated with the gene ontologies term in Supplementary Tables 20–23. For this analysis, we used the genome-wide background set and the following filters: fold change ≥1.3, number of peaks with motifs >1 and FDR $Q$-value ≤0.05. Finally, the resulting candidate TFs were checked for differential expression and only TFs with $Q$-value ≤0.05 were retained.

*Confocal and 3D-SIM super-resolution microscopy.* Following infection with BCG, THP-1 cells were seeded onto coverslips. mcherry-BCG-infected cells were loaded with 5 μM APG-4 (TEFLabs) with PowerLoad (Invitrogen), while GFP-BCG-infected cells were stained with LysoTracker Red DND-99 (Molecular Probes), according to the manufacturers' instructions. Cells were then fixed in 4% (v/v) formaldehyde, permeabilized with 0.3% (v/v) Triton X-100 in PBS and blocked with 5% (w/v) BSA in PBS. Cells were stained for Kir4.2 (Sigma), followed by secondary antibody conjugated with Alexa Fluor 647 (Molecular Probes) and Hoechst 33258 (Sigma). Stained cells were mounted with Vectashield H-1000 (Vector Laboratories). Confocal imaging was performed using a laser scanning microscope LSM 800 with Airyscan (Carl Zeiss). A Plan-Apochromat ×63/1.40 Oil DIC M27 objective lens and excitation wavelengths of 405, 488, 561 and 640 nm were used. The confocal pinhole was set to 1 Airy unit for the green channel and other channels adjusted to the same optical slice thickness. Airyscan processing was performed with ZEN software platform (Carl Zeiss). Three-dimensional-SIM images were acquired on a DeltaVision OMX v4 Blaze microscope (GE Healthcare) equipped with 405, 488, 561 and 647 nm lasers and a solid-state illuminator (WF) for excitation, and a BGR filter drawer (emission wavelengths 436/31 for DAPI, 528/48 for Alexa Fluor 488, 609/37 for Alexa Fluor 568, and 683/40 for Alexa Fluor 642). An Olympus Plan-Apochromat ×100/1.4 Point Spread Function (PSF) oil immersion objective lens was used with liquid-cooled Photometrics Evolve EM-CCD cameras (Photometrics) for each channel. Fifteen images per section per channel were acquired (made up of three rotations and five phase movements of the diffraction grating) at a z-spacing of 0.125 μm. Structured illumination deconvolution followed by alignment was carried out with SoftWorX (Applied Precision). Three-dimensional image reconstruction, figure and movie preparations were done with Imaris software (Andor-Bitplane).

*Mitosox red staining.* Cells ($1 \times 10^6$) were stained with 5 μM Mitosox red (ThermoFisher), diluted in PBS, for 30 min at RT in the dark. Cells were washed with PBS and stained with Vybrant DiD cell labelling solution (ThermoFisher) diluted in PBS (5 μl ml$^{-1}$ PBS). Cells were washed with PBS and mounted onto ibidi μ-Slide VI 0.1 slides precoated with poly-L-lysine (Merck) and visualized using an Olympus confocal microscope.

*Patch-clamp studies.* Whole-cell patch-clamp experiments were carried out on a QPatch-48 automated electrophysiology platform (Sophion Biosciences) using disposable 48-channel planar patch chip plates. Cell positioning and sealing parameters were set as follows: positioning pressure −70 mbar, minimum seal resistance 0.1 GΩ, holding potential −80 mV, holding pressure −20 mbar, minimum seal resistance for whole-cell requirement 0.001 GΩ. Access was obtained with the following sequence: (1) suction pulses in 29 mbar increments from −250 mbar to −453 mbar; (2) a suction ramp amplitude of −450 mbar; (3) −400 mV voltage zaps of 1 ms duration (10×). Following establishment of the whole-cell configuration, a ramp protocol of −120 to +120 mV in 200 ms was applied. Extracellular buffers of different potassium concentrations were exchanged from low to high without washing in between. Barium (0.1 mM) was added to the extracellular buffer containing 50 mM $[K^+]_o$. The external solutions (pH 7.4, 310 mOsm) contained (in mM): 160 NaCl, 10 HEPES, 4.5 KCl, 1 $MgCl_2$ and 2 $CaCl_2$ (4.5 mK $[K^+]_o$); 115 NaCl, 10 HEPES, 50 KCl, 1 $MgCl_2$ and 2 $CaCl_2$ (50 mK $[K^+]_o$); and 4.5 NaCl, 10 HEPES, 160 KCl, 1 $MgCl_2$ and 2 $CaCl_2$ (160 mK $[K^+]_o$). The internal solution (pH 7.2, 303 mOsm) contained (in mM): 120 KCl, 10 HEPES, 1.75 $MgCl_2$, 1 $Na_2ATP$, 10 EGTA and 8.6 $CaCl_2$.

*Generation of VSV.G pseudotyped lentiviral and transduction of human primary monocytes (KCNJ15++ cells).* An HIV-1-based lentiviral vector encoding the enhanced green fluorescent protein (EGFP) driven by Cytomegalovirus (CMV) or the constitutive EF1a promoter for the expression of the gene of interest in a SIN configuration was used. Custom synthesized (IDT Biotech) human *KCNJ15* complementary DNA (cDNA) was sub-cloned into the PE1A plasmid (Invitrogen, A10462), followed by gateway transfer of *KCNJ15* into the lentiviral vector. In brief, HEK293 cells were transfected using Xfect (TakaRa, 631318) with the viral packaging construct (pMDLg/pRRE (Addgene, 12251), pRSV/REV (Addgene, 12253), pMD2.G/V-SVG (Addgene, 8454)). Lentiviral particles were collected from the supernatant 48–72 h after transfection, concentrated by LentiX concentrator (Clontech, 631232) and titrated by quantitative PCR (qPCR) (determination of number of transducing or infectious units per ml) on HeLa cells. Titres for control and *KCNJ15* lentivirus were $6 \times 10^7$ TU ml⁻¹. Blood-derived CD14⁺ monocytes were adjusted to a concentration of $1 \times 10^6$ cells per ml; for gene transduction, duplicate wells of a 12-well plate were seeded with 1 ml per well of the cell suspension ($1 \times 10^6$ cells per ml) and virus was added in the presence of 6 μg ml⁻¹ polybrene (Sigma-Aldrich, 107689) (multiplicity of infection of 25). Cells were then incubated overnight with the virus and residual vector virus, non-adherent cells were removed by washing the wells with growth medium and the plate was further incubated at 37 °C. Two days after the initial addition of vector virus, transduction efficiency was determined by flow cytometry.

*Transfection of siRNAs.* THP-1 cells were transfected with siRNAs using the Lipofectamine2000 Kit (Invitrogen, 11668-019), according to the manufacturer's protocol. Knockdown was confirmed by reverse transcription qPCR (RT-qPCR) and western blotting.

*Western blot.* Protein lysates from cells were obtained by lysis in radioimmunoprecipitation assay buffer (Merck Millipore, R0278) with protease and phosphatase inhibitors (Roche, 04906837001). A Micro BCA Protein kit (ThermoFisher, 23235) was used to measure protein levels, and equal amounts of proteins were resolved by electrophoresis on 12% Tris-HCl gels (Mini-PROTEAN TGX gels; Bio-Rad, 4561043) and transferred onto polyvinylidene difluoride membranes (Trans-Blot Turbo Transfer Pack; Bio-Rad, 1704156). Membranes were developed using the indicated primary antibody at a 1:1,000 dilution and secondary antibodies at a 1:3,000 dilution in blocking solution. This was followed by incubation with chemiluminescent HRP detection reagent (Merck Millipore, WBKLS0500) for 1 min before image acquisition by the ChemiDoc Imaging System (Bio-rad).

*Annexin V and propidium iodide (PI) staining.* Cells ($0.5 \times 10^6$) were washed with PBS, followed by 1X Annexin V binding buffer. Cells were resuspended in 50 ul of binding buffer, added with 2.5 μl Annexin V APC (Biolegend, 640932) and incubated for 15 min at RT in the dark. Cells were then washed with binding buffer and resuspended in 100 μl binding buffer. PI (2.5 μl) was added and cells were acquired on an LSR Fortessa cytometer (BD Biosciences). Flow data were analysed with FlowJo software (Tree Star).

*RNA-sequencing (Cohort 3).* RNA-seq was performed on 39 granulocyte and 39 monocyte samples from ATB and HC individuals (Supplementary Tables 14 and 15). RNA-seq was also performed on KCNJ15⁺⁺ and control monocytes. Total RNA was extracted using the Ambion *mir*Vana miRNA Isolation Kit (Ambion ThermoFisher, AM1561), according to the manufacturer's protocol. All human RNAs were analysed on an Agilent Bioanalyzer for quality assessment. RNA QC: RNA integrity number ranged from 7.1 to 10, with a median value of 9.5. cDNA libraries were prepared using 300 ng total RNA and 2 ul of a 3:1,000 dilution of External RNA Control Consortium (ERCC) RNA spike-in controls (Ambion ThermoFisher, 4456739). The fragmented mRNA samples were subjected to cDNA synthesis using Illumina TruSeq RNA sample preparation kit version 2 (Illumina,

RS-122-2001 and RS-122-2002), according to the manufacturer's protocol, except for the following modifications: use of 12 PCR cycles and two additional rounds of Agencourt Ampure XP SPRI beads (Beckman Courter, A63881) to remove >600 bp double-stranded cDNA. The length distribution of the cDNA libraries was monitored using DNA 1000 kits on the Agilent bioanalyzer (Agilent). All samples were subjected to an indexed paired-end (PE) sequencing run of 2× 51 cycles on an Illumina HiSeq 2000 (Illumina) (3–4 samples per lane). FASTQ files were mapped to the human genome build hg19 using STAR. The mapped reads were then counted using featureCounts on the basis of GENCODE V19 annotations to generate gene counts.

*RNA-seq data analysis.* Gene-specific read counts were normalized to Reads Per Kilobase Million (RPKM) values and a consensus set of expressed genes was defined as those that were expressed ($\log_2(RPKM) \geq -0.5$) in at least 5 individuals. This yielded 12,465 expressed genes for granulocytes and 13,309 for monocytes. One granulocyte sample was discarded due to poor library quality (percent reads mapped, 45.3% and percent uniquely mapped reads, 43.1%). Age, sex, ethnicity and 4 technical covariates (number of reads, fraction of reads in exonic regions, percent of reads mapped to the human genome, percent uniquely mapped reads) were regressed out from the $\log_2(RPKM)$ values (Supplementary Fig. 3). DE genes were subsequently ascertained by the same two-step procedure as was used on ChIP-seq data (Supplementary Tables 16 and 17). As before, we used the 2-sample two-sided Wilcoxon test to quantify statistical significance (FDR $Q$-value $\leq 0.10$, fold change $\geq 1.5$). Differential expression was correlated with differential acetylation by associating DE genes with all DA peaks within 10 Kb of their transcription start site (Supplementary Table 18). Gene ontology analysis was performed using GOrilla[22] provided with a foreground (DE genes) and background (all expressed genes) list. Statistical significance was computed using Fisher's exact test (FDR $Q$-value $\leq 0.05$, fold change $\geq 1.5$) and redundant terms were defined as above (overlap of $\geq 40\%$ in the number of genes). The complete list of GOrilla results including redundant terms is shown in Supplementary Tables 29–31.

*Transcriptomic data.* KCNJ15 gene expression profiles were analysed in the 77 RNA-seq samples (Supplementary Tables 14 and 15) and four publicly available clinical datasets of whole-blood gene expression profiles from ATB patients ($n = 92$) and healthy individuals ($n = 61$), and patients with TB undergoing therapy at various time points ($n = 7$) (Supplementary Table 36). The following primers were used to measure KCNJ15 expression by RT-qPCR in in vitro experiments: F - CCCGGTGAGCCCATTTCAAATC and R - GACCAACTGAGCAACCAACAGG.

*Statistical significance of LD between haQTLs and GWAS SNPs.* We downloaded phenotype-associated SNPs from the NHGRI-EBI GWAS catalogue[47] (downloaded May 2018; Supplementary Tables 46, 47, 49 and 51). Only SNPs with genome-wide significance ($P < 5 \times 10^{-8}$) were retained and duplicates were removed. GWAS SNPs annotated with GRCh38 coordinates were lifted over to GRCh37. To create the set of TB-associated SNPs, GWAS results from two TB association studies not represented in the catalogue were added to the list[61,62]. Since LD with haQTLs was calculated using genotypes from 1000 Genomes data, we obtained the subset of SNPs genotyped in 1000 Genomes, resulting in 3 TB-associated SNPs in the Asian population (out of 3), 10 in the European population (out of 17) and 0 in the African population (out of 5). LD between these 13 GWAS SNPs and haQTLs was calculated using 1000 Genomes data from the population in which the GWAS study was performed ($R^2 \geq 0.8$, distance $\leq 500$ kb). We next downloaded 30 SNPs associated with leprosy from NHGRI-EBI GWAS catalogue[47], all of which were in East Asian populations and also genotyped in 1000 Genomes. We obtained inflammation-associated SNPs from the GWAS catalogue[47], and found that 80 of the 81 EUR SNPs were genotyped in 1000 Genomes. We also used a set of 627 EUR autoimmune GWAS SNPs[40] with $P \leq 5 \times 10^{-8}$ (Supplementary Tables 46–52). The 69 granulocyte haQTLs and 83 monocyte haQTLs that are in LD with immune phenotypes are the unique haQTLs from Supplementary Tables 48, 50 and 52. To calculate the statistical significance of linkage between GWAS SNP sets and haQTLs (leprosy, autoimmune and inflammation; Fig. 6e,f), we used a method that corrects for allele frequency, distance to the nearest transcription start site and biases in LD block size between haQTL and control SNPs[40].

**Reporting Summary.** Further information on research design is available in the Nature Research Reporting Summary linked to this article.

## Data availability

ChIP-seq data have been deposited at the European Genome-phenome Archive EGA (http://www.ebi.ac.uk/ega/), which is hosted by the EBI, under accession number EGAS00001003118. RNA-seq data have been deposited at NCBI's Gene Expression Omnibus through GEO Series accession number GSE126614. Source data are provided with this paper.

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

## Acknowledgements

We thank the personnel of the Biological Resource Centre's BSL3 laboratory and the Defence Science Organization's BSL3 laboratory for facilitating this study. This project was supported by core funds from Singapore's Agency for Science, Technology and Research (A*STAR); A*STAR translational programme in infectious disease no. IAF11003; A*STAR Joint Council Office Grant No. JCO-CDA15302FG151; BMRC-SERC grant no. 1121480006; SIgN Immunomonitoring platform grant no. IAF311006; SIgN

and ID Labs core fund; BMRC transition fund grant no. H16/99/b0/011; Swiss National Foundation grant no. 310030-173240; the European Union's Research and Innovation Program grant no. TBVAC2020 643381; and Nanyang Technological University Singapore's Lee Kong Chian School of Medicine Start Up Grant. R.J.W. was supported by the Francis Crick Institute, which is funded by Cancer Research UK (FC0012018), MRC (UK) (FC0010218) and Wellcome (FC0010218). He also received support from Wellcome (104803, 203135) and NIH V01AI115940. For the purposes of open access, the authors have applied a CC-BY public copyright to any author-accepted manuscript arising from this submission.

## Author contributions

S.P. and A.S. conceived the idea. R.C.H.d.R., J.P., G.D.L., A.S. and S.P. designed the study and analysed the data. R.C.H.d.R., J.P., P.K., C.Y.C., C.L., C.R., G.C.W., N.A.R., Z.Z., J.L., B.L., F.Z., M.P., S.T.O., H.S.H., M.M., X.L. and A.L. performed the experiments. S.G. performed heritability analyses. K.G.C. designed and analysed the electrophysiological and APG-4 data. D.K. and O.R. contributed to overexpression experiments. C.B.E.C., Y.T.W., E.D.B. and R.J.W. provided clinical samples and interpreted the results. C.C.K. designed and performed array genotyping. J.P., R.C.H.d.R., A.S. and S.P. wrote the manuscript. All authors discussed results and contributed to the manuscript.

## Competing interests

The authors declare no competing interests.

## Additional information

**Extended data** is available for this paper at https://doi.org/10.1038/s41564-021-01049-w.

**Correspondence and requests for materials** should be addressed to Amit Singhal or Shyam Prabhakar.

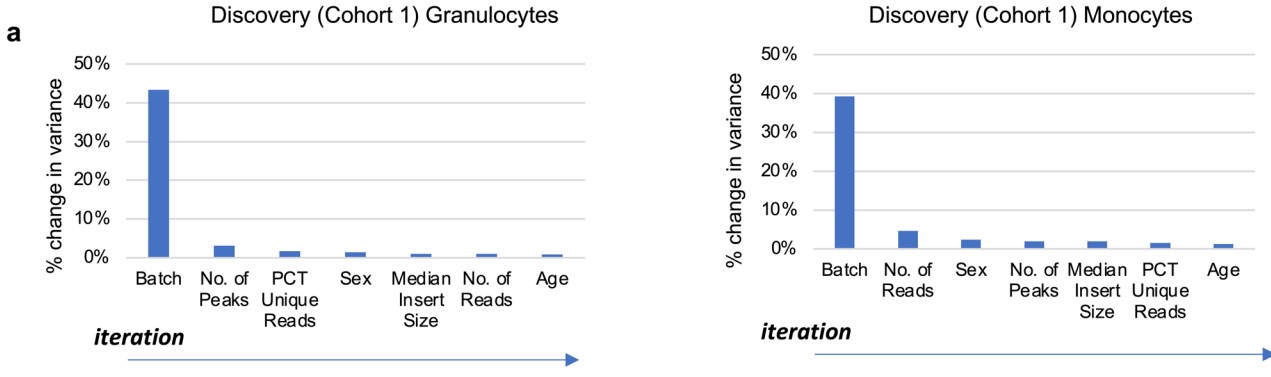

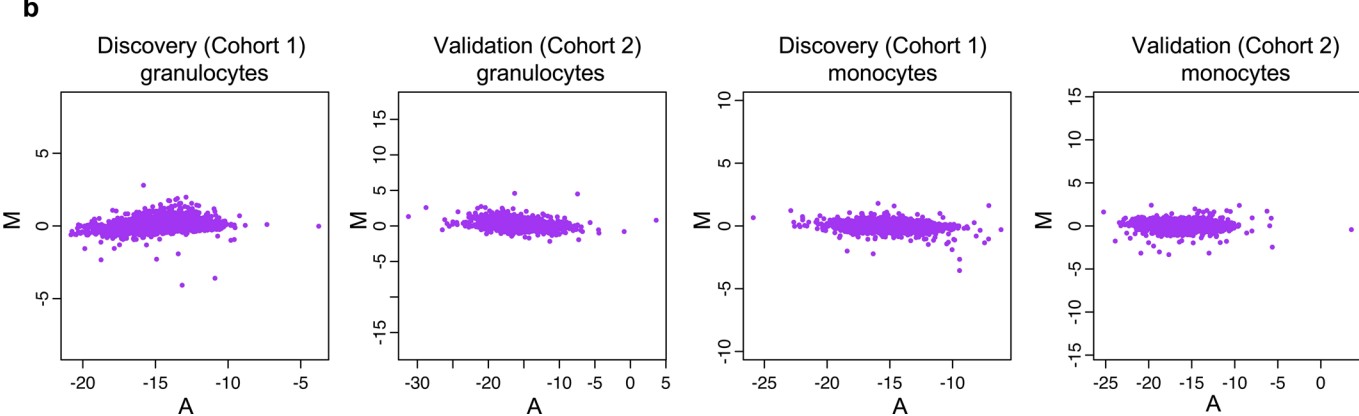

**Extended Data Fig. 1 | Regression of confounding variables in H3K27ac ChIP-seq data.** (**a**) Method for selecting covariates to be regressed in discovery samples. The covariate that explained the most variance at each iteration of the method was plotted. (**b**) MA plot of peak heights for the discovery and validation cohorts. Only consistent samples were used.

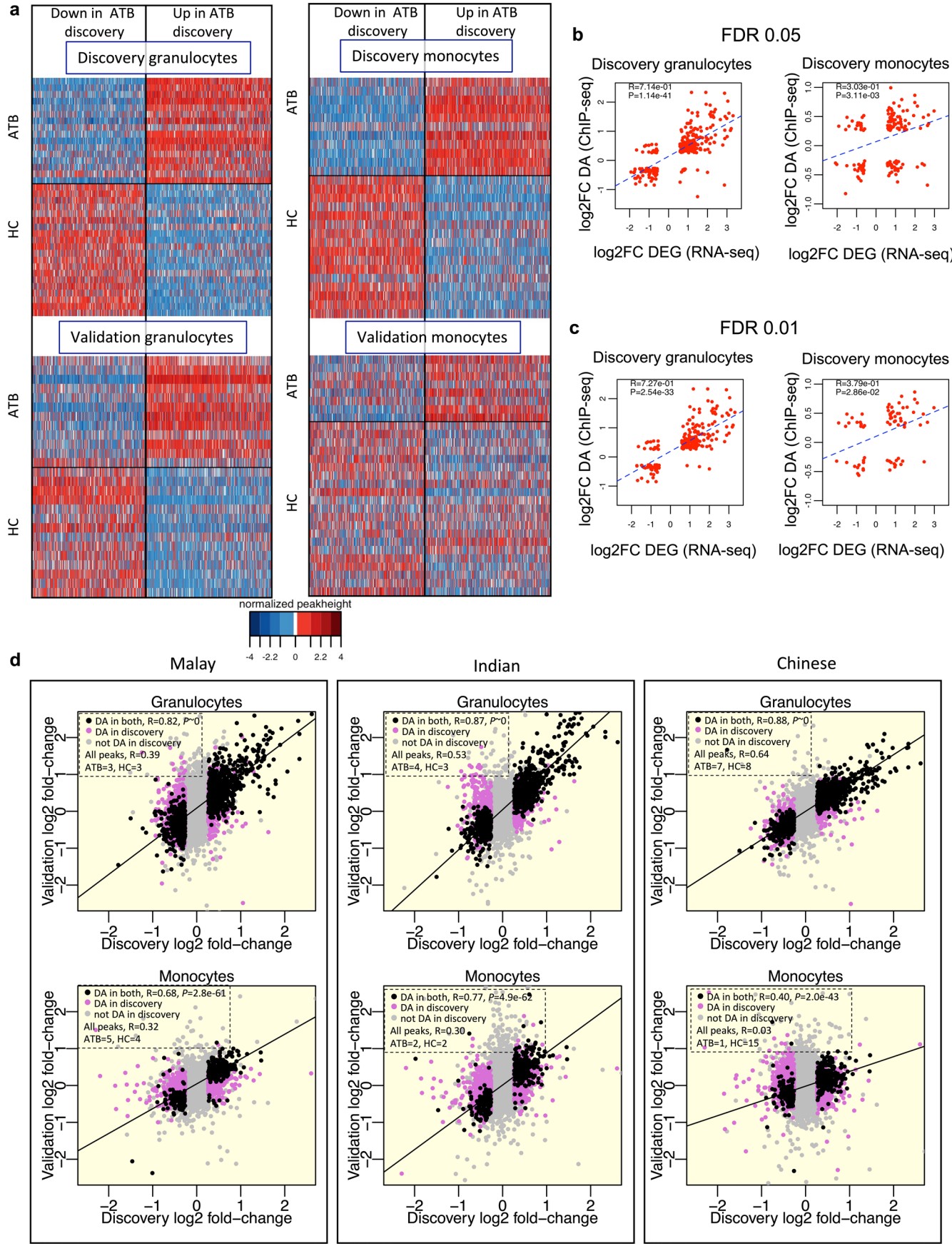

**Extended Data Fig. 2 | See next page for caption.**

**Extended Data Fig. 2 | ChIP-seq peak height z-scores, fold change correlation of ChIP-seq and RNA-seq data, and ethnicity-specific fold-change correlation for ChIP-seq data.** (**a**) DA peak height z-scores for the four ATB vs. HC Singapore (discovery and validation) datasets from samples from the core set. Each row represents a single individual, and each column a DA peak ascertained from the discovery cohort. The peaks in the top and bottom panels are arranged in the same order. Only consistent samples are displayed (granulocytes discovery: ATB = 16, HC = 20; monocytes discovery: ATB = 11, HC = 16; granulocytes validation: ATB = 12, HC = 14; monocytes validation: ATB = 8, HC = 21). (**b,c**) Correlation between differential acetylation and differential gene expression (related to Supplementary Table 18). DE and DA gene sets were defined using the default FDR threshold of 0.05 (b) as well as a more stringent FDR threshold of 0.01 (c). DA peaks were associated with DE genes using the GREAT tool and the Pearson correlation of log-fold change was calculated. P-values indicate concordance of the fold-change direction, calculated using a one-sided hypergeometric test. (**d**) Ethnicity-specific fold-change correlation between discovery and validation cohorts. Black dots are DA both in discovery and validation cohorts. Pink dots are DA only in the discovery cohort. R indicates Pearson correlation coefficient of log2(fold-change) between discovery and validation. *P*-value of concordance in fold-change direction for shared DA peaks was calculated using two-sided Fisher's exact test.

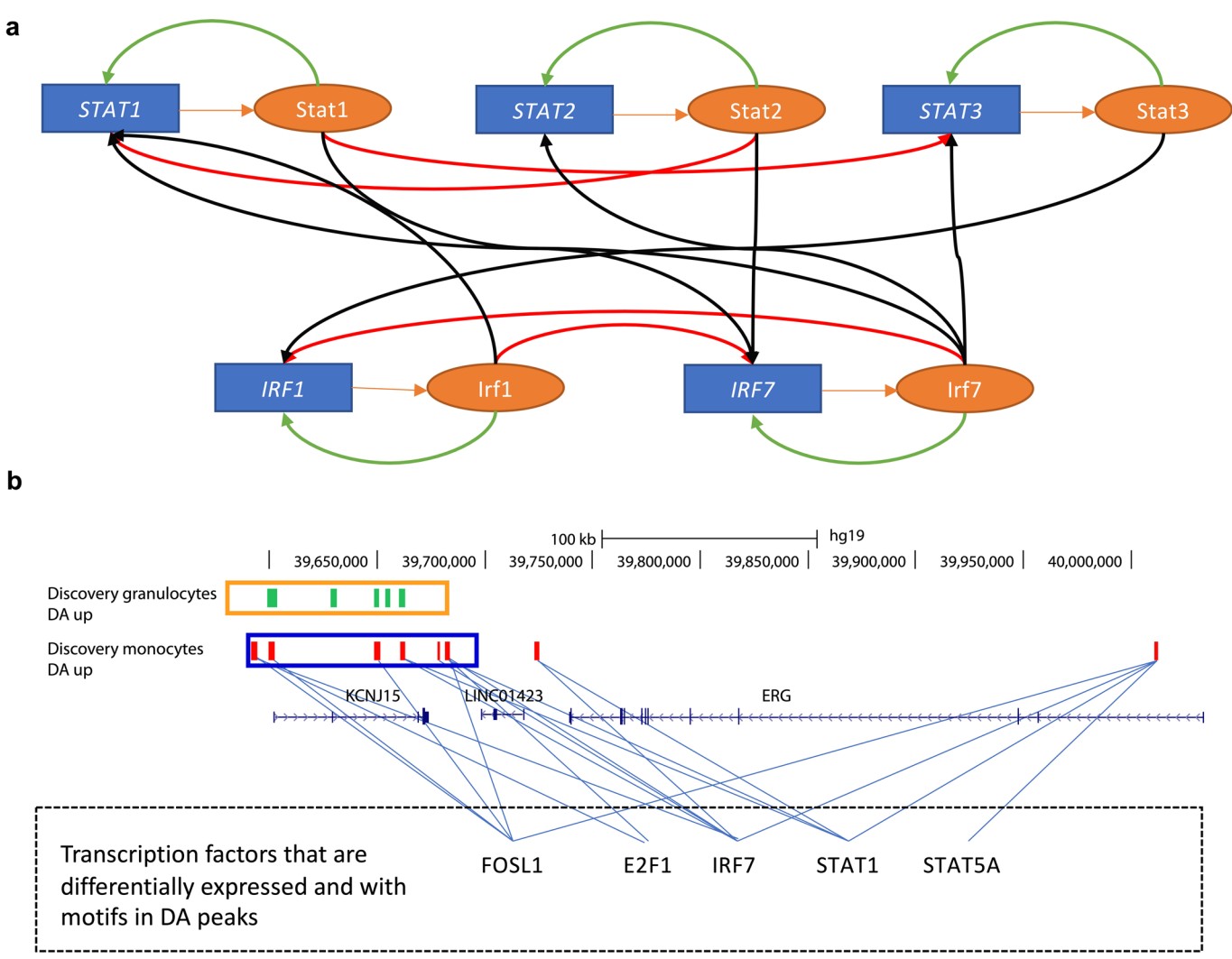

**Extended Data Fig. 3 | Transcription factors and super enhancers in the *KCNJ15* locus.** (**a**) Predicted gene regulatory interactions in granulocytes between differentially expressed TFs associated with DA peaks belonging to enriched GO terms in Fig. 5a. Blue boxes: genes. Orange ovals: proteins. Orange arrows: translation. Green arrows: autoregulatory loops. Red arrows: cross-regulation of paralogous TFs. Black arrows: cross-regulation of other TFs. (**b**) *KCNJ15* locus: green and red ticks indicate DA peaks up-regulated in response to infection in granulocytes and monocytes respectively. Orange and blue boxes indicate the corresponding super-enhancer regions. Blue lines: predicted transcription factor binding sites for master transcription factors in DA peaks.

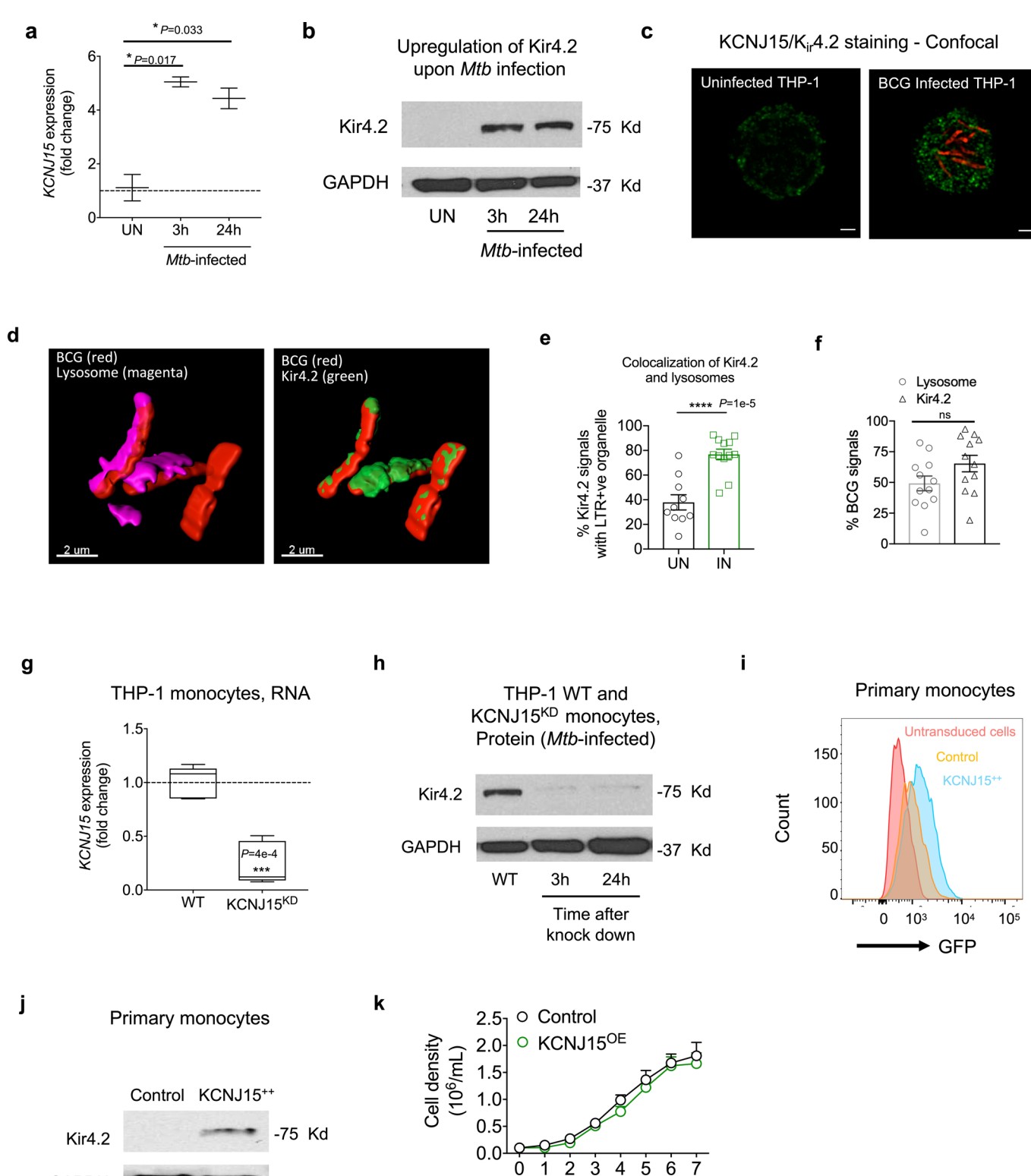

**a** $KCNJ15$ expression (fold change)
UN, 3h, 24h *Mtb*-infected
* *P*=0.017, * *P*=0.033

**b** Upregulation of Kir4.2 upon *Mtb* infection
Kir4.2 -75 Kd
GAPDH -37 Kd
UN, 3h, 24h *Mtb*-infected

**c** KCNJ15/K$_{ir}$4.2 staining - Confocal
Uninfected THP-1 | BCG Infected THP-1

**d** BCG (red) Lysosome (magenta) 2 um | BCG (red) Kir4.2 (green) 2 um

**e** Colocalization of Kir4.2 and lysosomes
% Kir4.2 signals with LTR+ve organelle
UN, IN
**** *P*=1e-5

**f** % BCG signals
○ Lysosome △ Kir4.2
ns

**g** THP-1 monocytes, RNA
$KCNJ15$ expression (fold change)
WT, KCNJ15$^{KD}$
*P*=4e-4 ***

**h** THP-1 WT and KCNJ15$^{KD}$ monocytes, Protein (*Mtb*-infected)
Kir4.2 -75 Kd
GAPDH -37 Kd
WT, 3h, 24h
Time after knock down

**i** Primary monocytes
Count, GFP
Untransduced cells, Control, KCNJ15$^{++}$

**j** Primary monocytes
Control, KCNJ15$^{++}$
Kir4.2 -75 Kd
GAPDH -37 Kd

**k** Cell density (10$^6$/mL)
○ Control ○ KCNJ15$^{OE}$
Days in culture

**Extended Data Fig. 4 | See next page for caption.**

**Extended Data Fig. 4 | Role of *KCNJ15*/Kir4.2 during Mycobacterial infection.** (a) *KCNJ15* mRNA was assessed by qRT-PCR in *Mtb*-infected primary monocytes. *KCNJ15* expression normalized to *GAPDH*, relative to uninfected (UN) cells, is shown. *P*-values from two-sided, unpaired *t*-test. Data from 2 independent experiments. (b) Western blot analysis of Kir4.2 and GAPDH in *Mtb*-infected primary monocytes as in a. Data from 2 independent experiments. (c) Immunostaining (confocal microscopy) of Kir4.2 in THP-1 cells, which was increased upon *M. bovis* BCG infection. Green: Kir4.2; red: mCherry- BCG. Scale-bars: 2μm. Data from 3 independent experiments. (d) Uninfected or mcherry-BCG infected THP-1 monocytes stained for endo-lysosomes with Lysotracker (LTR) and *KCNJ15*/Kir4.2, 24 h post-infection. Images from BCG-infected cells are shown. Scale bar 2 μm. Data from 2 independent experiments. (e) Compiled data of Kir4.2 co-localization with LTR in uninfected (UN) and BCG-infected (IN) cells, as shown in d. n = 10-12 cells; *P*-value from two-sided, unpaired *t*-test. (f) Percentage of BCG co-localized with either endo-lysosomes only or Kir4.2 only, from d. n = 10-12 cells. (g) siRNA-mediated knockdown of *KCNJ15* in THP-1 cells (KCNJ15[KD]), RT-PCR of *KCNJ15* gene. Data from 5 independent experiments *P*-value from two-tailed, unpaired *t*-test. (h) Western blot for Kir4.2 protein in KCNJ15[KD] THP-1 cells. Data from 2 independent experiments. (i) CD14[+] primary monocytes were transduced with either lentiviral plasmid carrying human *KCNJ15* cDNA or the control plasmid. Transduction efficiency was determined by FACS using GFP which is constitutively expressed. Pink: un-transduced cells; orange: cells transduced with empty control vector; blue: cells transduced with *KCNJ15* plasmid. Data from 3 independent experiments. (j) Transduction efficiency of primary monocytes in (i) was determined by Western blotting. Data from 2 independent experiments. (k) Growth curve of KCNJ15[OE] and Control cells (Control[OE]) *in vitro* in RPMI medium, indicating no defect in the growth of KCNJ15[OE] cells. Cells were seeded at 10[5] cells/ml. Data from 2 independent experiments. Data in a and g: box and whiskers, minimum to maximum. Data in e,f,k: mean±SEM. Statistically significant *P*-values: * P <= 0.05, ** P <= 0.01, ***P <= 0.001, **** P <= 0.0001.

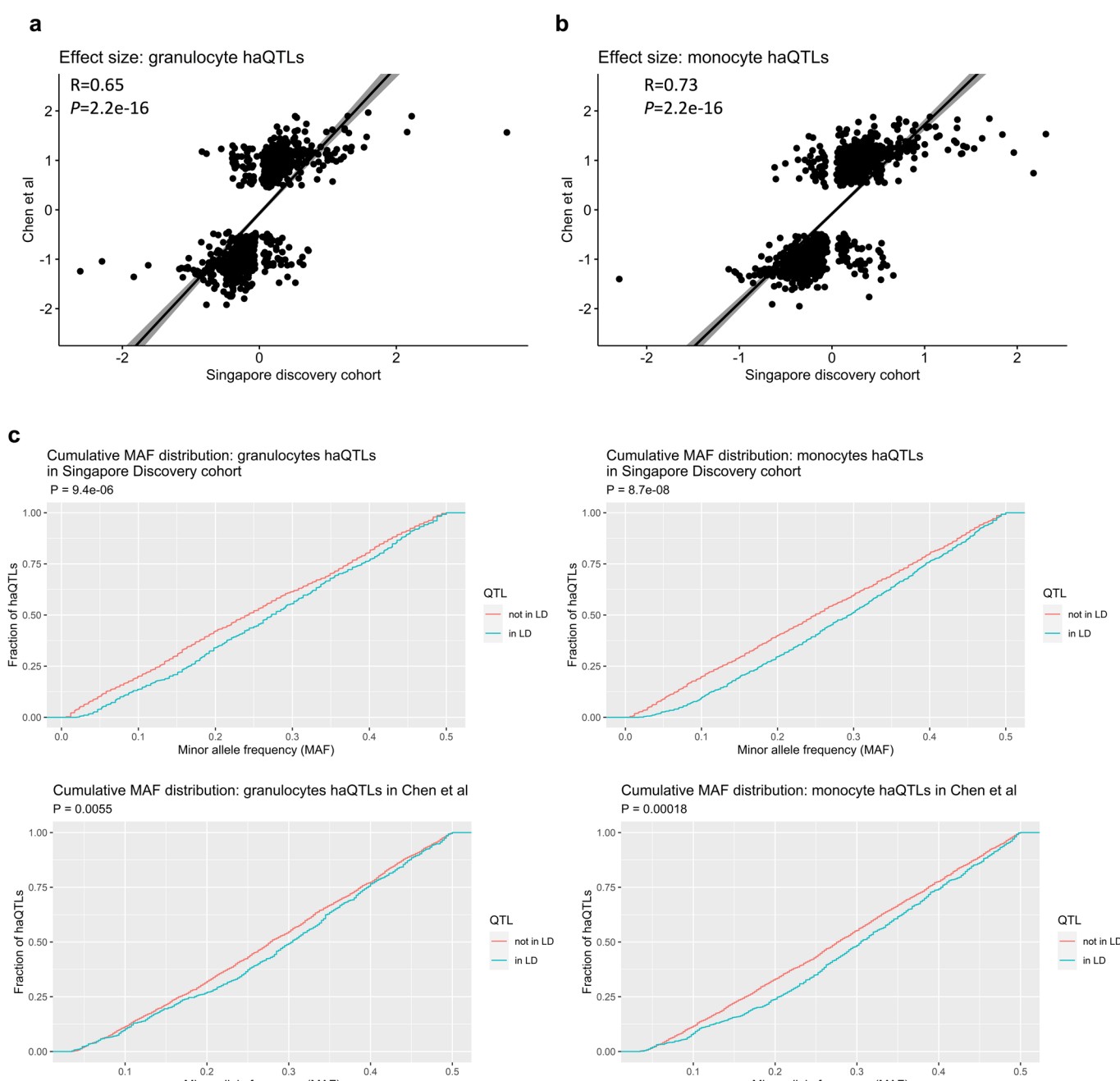

**Extended Data Fig. 5 | Comparison of haQTLs from different studies.** Comparison of granulocyte (**a**) and monocyte (**b**) haQTL effect size between the Singapore discovery cohort (Chinese) and the European cohort[19]. Each dot represents an haQTL in the Singapore discovery cohort (x-axis) and a corresponding European[19] haQTL in LD with the former. Effect size is defined as the regression coefficient *beta* relating ChIP-seq peak height to genotype. *R*: Pearson correlation. *P*-value: two-sided Pearson correlation *P*-value. (**c**) Cumulative minor allele frequency distributions for granulocyte and monocyte haQTLs in the Singapore discovery cohort (Chinese) and the European dataset[19]. Blue curves: shared haQTLs. Red curves: cohort-specific haQTLs. The statistical significance (Kolmogorov-Smirnov test, two-sample, two-sided) of the shift in allele frequency between shared and non-shared haQTLs is indicated in each panel.

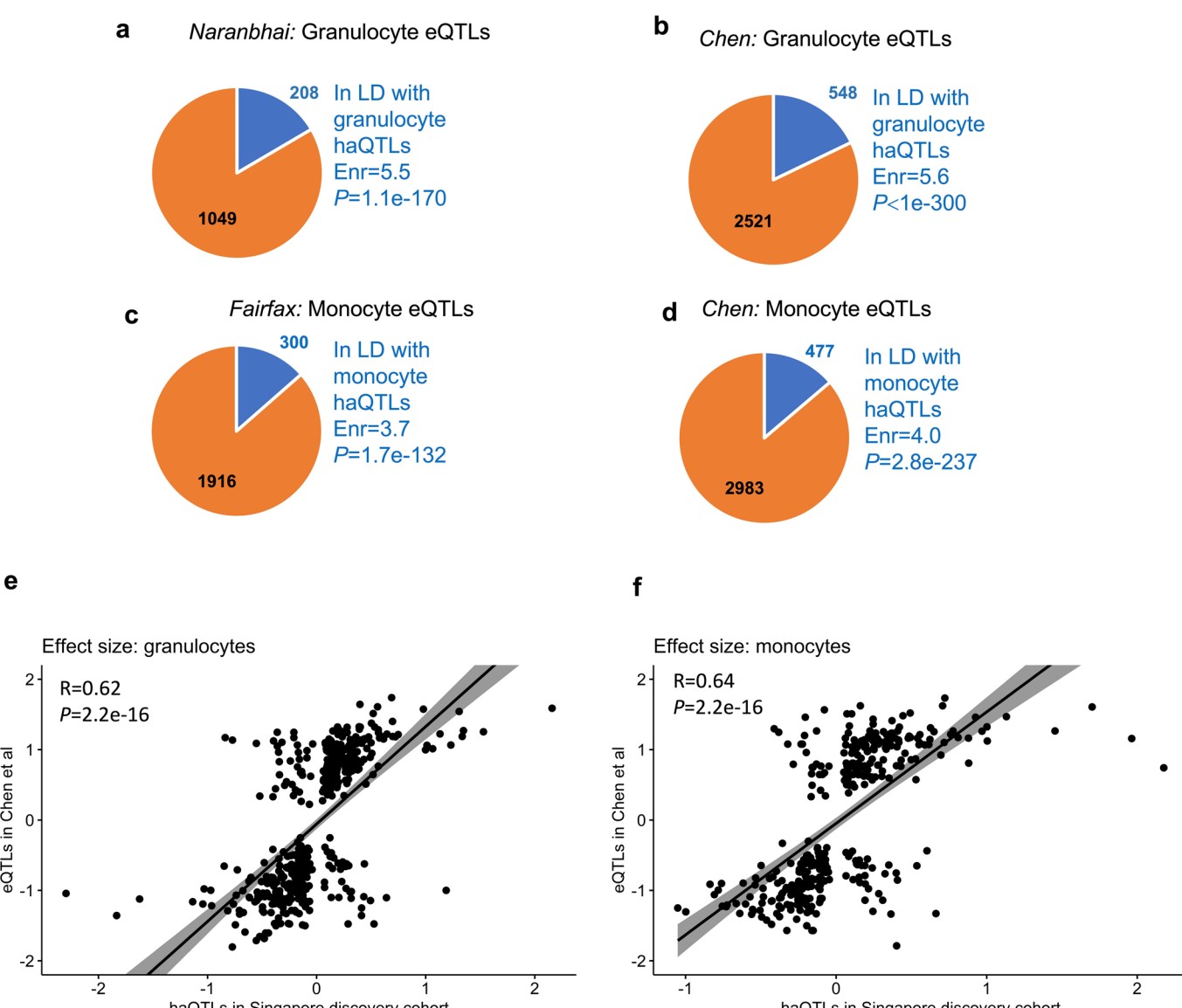

**Extended Data Fig. 6 | Comparison of haQTLs with eQTLs.** (**a**) – (**d**) Pie charts of granulocyte and monocyte eQTLs from previous studies that are in LD with corresponding haQTLs from this study. Enr: fold-enrichment; *P*-value: Z-score test (details are in Methods section on "Statistical significance of LD between haQTLs and eQTLs"). (**e**,**f**) Comparison of effect size between haQTLs in Singapore discovery cohort and eQTLs in the European dataset[19] for (e) granulocytes and (f) monocytes. Each dot represents an haQTL in the Singapore discovery cohort (x-axis) and a corresponding eQTL in the European[19] dataset that is in LD with the former. Effect size is defined as the regression coefficient *beta* relating ChIP-seq peak height to genotype. *R*: Pearson correlation. *P*-value: two-sided Pearson correlation *P*-value.

# nature research

# Reporting Summary

Nature Research wishes to improve the reproducibility of the work that we publish. This form provides structure for consistency and transparency in reporting. For further information on Nature Research policies, see Authors & Referees and the Editorial Policy Checklist.

## Statistics

For all statistical analyses, confirm that the following items are present in the figure legend, table legend, main text, or Methods section.

| n/a | Confirmed | |
|---|---|---|
| ☐ | ☒ | The exact sample size (*n*) for each experimental group/condition, given as a discrete number and unit of measurement |
| ☐ | ☒ | A statement on whether measurements were taken from distinct samples or whether the same sample was measured repeatedly |
| ☐ | ☒ | The statistical test(s) used AND whether they are one- or two-sided <br> *Only common tests should be described solely by name; describe more complex techniques in the Methods section.* |
| ☐ | ☒ | A description of all covariates tested |
| ☐ | ☒ | A description of any assumptions or corrections, such as tests of normality and adjustment for multiple comparisons |
| ☐ | ☒ | A full description of the statistical parameters including central tendency (e.g. means) or other basic estimates (e.g. regression coefficient) AND variation (e.g. standard deviation) or associated estimates of uncertainty (e.g. confidence intervals) |
| ☐ | ☒ | For null hypothesis testing, the test statistic (e.g. *F*, *t*, *r*) with confidence intervals, effect sizes, degrees of freedom and *P* value noted <br> *Give P values as exact values whenever suitable.* |
| ☒ | ☐ | For Bayesian analysis, information on the choice of priors and Markov chain Monte Carlo settings |
| ☒ | ☐ | For hierarchical and complex designs, identification of the appropriate level for tests and full reporting of outcomes |
| ☐ | ☒ | Estimates of effect sizes (e.g. Cohen's *d*, Pearson's *r*), indicating how they were calculated |

*Our web collection on statistics for biologists contains articles on many of the points above.*

## Software and code

Policy information about availability of computer code

| Data collection | No software was used for data collection |
|---|---|
| Data analysis | SAMtools (Li, 2011) version 0.1.19-44428cd <br> BWA (Li and Durbin, 2009) version 0.7.10-r789 <br> Picard Tools broadinstitute.github.io/picard/ <br> BedTools (Quinland and Hall 2010) version 2.25.0 <br> DFilter (Kumar et al., 2013) version 1.6 http://collaborations.gis.a-star.edu.sg/~cmb6/kumarv1/dfilter/ <br> GREAT (McLean et al., 2010) version 3.0.0 <br> GORilla (Eden et al., 2009) http://cbl-gorilla.cs.technion.ac.il <br> HOMER (Heinz et al., 2010) version 4.10 <br> GATK version 3.2-2 (DePristo et al., 2011) https://software.broadinstitute.org/gatk/ <br> G-SCI test (del Rosario et al., 2015) http://collaborations.gis.a-star.edu.sg/~cmb6/G-SCI_test/ <br> R The R Project for Statistical Computing https://www.r-project.org/ <br> UCSC Genome Browser University of California Santa Cruz https://genome.ucsc.edu/ <br> FlowJo Three Star https://www.flowjo.com/ <br> ZEN software platform  Carl Zeiss https://www.zeiss.com/microscopy/int/products/microscope-software/zen.html <br> softWoRX Suite 2.0 Applied Precision, GE Healthcare http://www.gelifesciences.co.kr/wp-content/uploads/2016/11/softworx_suite_overview.pdf <br> Imaris software (Andor-Bitplane, Zurich) Andor-Bitplane, Zurich www.bitplane.com/ <br> LD score regression (Finucane, et al., 2015) https://github.com/bulik/ldsc |

For manuscripts utilizing custom algorithms or software that are central to the research but not yet described in published literature, software must be made available to editors/reviewers. We strongly encourage code deposition in a community repository (e.g. GitHub). See the Nature Research guidelines for submitting code & software for further information.

## Data

Policy information about availability of data

All manuscripts must include a data availability statement. This statement should provide the following information, where applicable:
- Accession codes, unique identifiers, or web links for publicly available datasets
- A list of figures that have associated raw data
- A description of any restrictions on data availability

ChIP-seq data have been deposited at the European Genome-phenome Archive EGA, http://www.ebi.ac.uk/ega/), which is hosted by the EBI, under accession number EGAS00001003118. RNA-seq data have been deposited at NCBI's Gene Expression Omnibus through GEO Series accession number GSE126614. The following figures have associated raw data provided as excel files: Figs. 1-4 and 6.

# Field-specific reporting

Please select the one below that is the best fit for your research. If you are not sure, read the appropriate sections before making your selection.

☒ Life sciences    ☐ Behavioural & social sciences    ☐ Ecological, evolutionary & environmental sciences

For a reference copy of the document with all sections, see nature.com/documents/nr-reporting-summary-flat.pdf

# Life sciences study design

All studies must disclose on these points even when the disclosure is negative.

| | |
|---|---|
| Sample size | 238 ChIP-seq, 78 RNA-seq |
| Data exclusions | For ChIP-seq samples, we discarded samples with low complexity as measured by the number of unique read-pairs (<15 million for Batches 1 and 2; <30 million for Batch 3). We also discarded samples with high GC bias (>4,500 peaks showing greater than two-fold GC bias in either direction). For RNA-seq samples, we discarded one sample with poor quality (percent reads mapped =45.3% and percent uniquely mapped reads=43.1%). |
| Replication | To ensure replication we performed a replication study (for ChIP-seq only) |
| Randomization | Covariates were controlled for by regression (see methods) |
| Blinding | The study was undertaken to find the H3K27ac and transcriptomic differences between known tuberculosis and healthy individuals. Hence no blinding was performed |

# Reporting for specific materials, systems and methods

We require information from authors about some types of materials, experimental systems and methods used in many studies. Here, indicate whether each material, system or method listed is relevant to your study. If you are not sure if a list item applies to your research, read the appropriate section before selecting a response.

## Materials & experimental systems

| n/a | Involved in the study |
|---|---|
| ☐ | ☒ Antibodies |
| ☐ | ☒ Eukaryotic cell lines |
| ☒ | ☐ Palaeontology |
| ☒ | ☐ Animals and other organisms |
| ☐ | ☒ Human research participants |
| ☒ | ☐ Clinical data |

## Methods

| n/a | Involved in the study |
|---|---|
| ☐ | ☒ ChIP-seq |
| ☐ | ☒ Flow cytometry |
| ☒ | ☐ MRI-based neuroimaging |

## Antibodies

Antibodies used
Anti-KCNJ15 (Sigma-Aldrich, #HPA016702)
anti-APAF-1 (Cell Signaling Technology, #5088)
 anti-beta-actin (Cell Signaling Technology, #4967)
Annexin V APC (Biolegend, #640932)
anti–glyceraldehyde-3-phosphate dehydrogenase (GAPDH) (14C10) (Cell Signaling Technology, #2118)
anti-rabbit immunoglobulin G (IgG) horseradish peroxidase (HRP)-linked antibody (Cell Signaling Technology, #7074)
mouse IgG1 kappa isotype control eFluor 450 (eBioscience, #48-4714-82)
H3K27ac antibody for all ChIP experiments (Catalogue #39133; Active Motif)

| Validation | Each antibody is validated for FACS staining as per manufacturers description. We also used FMO and isotype controls for validating the staining. |

## Eukaryotic cell lines

Policy information about cell lines

| Cell line source(s) | Human monocyte THP-1 cells from ATTC |
| Authentication | This cell line was bought from ATCC |
| Mycoplasma contamination | This cell line was tested for Mycoplasma and was found to be negative. |
| Commonly misidentified lines (See ICLAC register) | *Name any commonly misidentified cell lines used in the study and provide a rationale for their use.* |

## Human research participants

Policy information about studies involving human research participants

| Population characteristics | See Supplementary Table 1 and Supplementary Table 19 |
| Recruitment | Singaporean Cohort (Cohorts 1-3):<br>HIV-negative ATB patients (based on clinical diagnosis with mycobacterial and radiographic evidence) and HC individuals (negative for IFN-γ release assay (IGRA); QuantiFERON TB gold test) were recruited at Tan Tock Seng Hospital's Tuberculosis Control Unit (TTSH, TBCU). Additional HC individuals were recruited locally at SIgN. Patients were sampled within 4 days of anti-TB treatment initiation and excluded if they had previously received anti-TB therapy. All participants provided written informed consent.<br><br>South African Cohort (Cohort 4):<br>1) ATB: All ATB patients (median age: 27.3; interquartile range (IQR): 23-33; 60% male) were tested sputum Xpert MTB/RIF (Xpert, Cepheid, Sunnyvale, CA) positive and had clinical symptoms and/or radiographic evidence of TB. All ATB cases were drug sensitive and had received no more than one dose of anti-tubercular treatment at the time of baseline blood sampling. A follow-up sample was obtained at completion of TB treatment (24 weeks).<br>2) LTBI: All individuals with latent Mtb infection (median age: 26.5; IQR: 23-33; 60% male) were asymptomatic, had a positive IFN-γ release assay (IGRA, QuantiFERON®-TB Gold In-Tube, Qiagen, Hilden), tested sputum Xpert MTB/RIF negative and exhibited no clinical evidence of active TB.<br>3) HC: Health controls (medium age: 30.5; IQR: 24-32; 40% male) were negative for IFN-γ release assay, tested sputum Xpert MTB/RIF negative and exhibited no clinical evidence of active TB.<br><br>In all cohorts, no compensation was given to any participant. |
| Ethics oversight | This study was approved by the Domain Specific Review Board of the National Healthcare Group (#2010/00566) and Institutional Review Board of the National University of Singapore (#09-256).<br>The South African cohort (Cohort 4) was recruited from the Ubuntu Clinic, Site B, Khayelitsha (Cape Town, South Africa) between March 2017 and December 2018. All participants were HIV-uninfected adults (age ≥ 18 yr) and provided written informed consent. The study was approved by the University of Cape Town Human Research Ethics Committee (HREC: 050/2015) and was conducted under DMID protocol no.15-0047. |

Note that full information on the approval of the study protocol must also be provided in the manuscript.

## ChIP-seq

### Data deposition

☒ Confirm that both raw and final processed data have been deposited in a public database such as GEO.

☒ Confirm that you have deposited or provided access to graph files (e.g. BED files) for the called peaks.

| Data access links<br>*May remain private before publication.* | EGAS00001003118 (EGA) |
| Files in database submission | T1G3_CON_dedupped.bam<br>T1G4_CON_dedupped.bam<br>T1G5_CON_dedupped.bam<br>T2G1_INF_dedupped.bam<br>T2G2_CON_dedupped.bam<br>T2G4_CON_dedupped.bam<br>T2G5_INF_dedupped.bam<br>T3G2_INF_dedupped.bam<br>T3G5_INF_dedupped.bam<br>T3G6_INF_dedupped.bam |

```
T4G2_INF_dedupped.bam
T4G5_CON_dedupped.bam
T9G1_INF_dedupped.bam
T10G1_CON_dedupped.bam
T10G3_INF_dedupped.bam
T10G4_INF_dedupped.bam
T11G1_CON_dedupped.bam
T11G2_INF_dedupped.bam
T11G3_CON_dedupped.bam
T11G4_CON_dedupped.bam
T12G4_INF_dedupped.bam
T12G6_INF_dedupped.bam
T13G3_INF_dedupped.bam
T13G5_INF_dedupped.bam
T13G6_INF_dedupped.bam
T14G1_CON_dedupped.bam
T14G4_INF_dedupped.bam
CHP049_CON_dedupped.bam
CHP050_CON_dedupped.bam
CHP052_INF_dedupped.bam
CHP053_INF_dedupped.bam
CHP055_INF_dedupped.bam
CHP056_CON_dedupped.bam
CHP057_INF_dedupped.bam
CHP058_CON_dedupped.bam
CHP059_CON_dedupped.bam
CHP091_CON_dedupped.bam
CHP092_INF_dedupped.bam
CHP094_CON_dedupped.bam
CHP095_CON_dedupped.bam
CHP097_CON_dedupped.bam
CHP098_INF_dedupped.bam
CHP099_CON_dedupped.bam
CHP100_CON_dedupped.bam
CHP101_CON_dedupped.bam
T1G1_INF_dedupped.bam
T1G6_CON_dedupped.bam
T2G6_INF_dedupped.bam
T3G1_CON_dedupped.bam
T3G4_INF_dedupped.bam
T4G3_INF_dedupped.bam
T4G4_INF_dedupped.bam
T4G6_CON_dedupped.bam
T9G2_CON_dedupped.bam
T9G3_CON_dedupped.bam
T9G4_CON_dedupped.bam
T9G5_INF_dedupped.bam
T10G2_CON_dedupped.bam
T10G5_CON_dedupped.bam
T10G6_CON_dedupped.bam
T11G5_INF_dedupped.bam
T11G6_CON_dedupped.bam
T12G1_INF_dedupped.bam
T13G2_INF_dedupped.bam
T13G4_INF_dedupped.bam
T14G3_CON_dedupped.bam
CHP048_CON_dedupped.bam
CHP051_INF_dedupped.bam
CHP054_INF_dedupped.bam
CHP090_INF_dedupped.bam
CHP096_CON_dedupped.bam
CHP118_CON_dedupped.bam
CHP119_INF_dedupped.bam
T5M3_CON_dedupped.bam
T5M4_INF_dedupped.bam
T5M5_CON_dedupped.bam
T6M1_INF_dedupped.bam
T6M2_CON_dedupped.bam
T6M4_INF_dedupped.bam
T7M2_INF_dedupped.bam
T7M5_INF_dedupped.bam
T8M1_INF_dedupped.bam
T8M2_INF_dedupped.bam
T8M4_CON_dedupped.bam
T8M5_INF_dedupped.bam
```

```
T8M6_INF_dedupped.bam
CHP060_CON_dedupped.bam
CHP064_CON_dedupped.bam
CHP067_CON_dedupped.bam
CHP068_INF_dedupped.bam
CHP069_INF_dedupped.bam
CHP070_CON_dedupped.bam
CHP071_CON_dedupped.bam
CHP072_INF_dedupped.bam
CHP073_INF_dedupped.bam
CHP074_CON_dedupped.bam
CHP076_INF_dedupped.bam
CHP077_CON_dedupped.bam
CHP079_CON_dedupped.bam
CHP080_INF_dedupped.bam
CHP081_INF_dedupped.bam
CHP082_CON_dedupped.bam
CHP087_CON_dedupped.bam
CHP088_CON_dedupped.bam
CHP108_CON_dedupped.bam
T5M2_CON_dedupped.bam
T5M6_CON_dedupped.bam
T6M3_INF_dedupped.bam
T6M5_CON_dedupped.bam
T6M6_CON_dedupped.bam
T7M1_CON_dedupped.bam
T7M4_INF_dedupped.bam
T7M6_INF_dedupped.bam
T8M3_CON_dedupped.bam
CHP061_CON_dedupped.bam
CHP065_INF_dedupped.bam
CHP066_INF_dedupped.bam
CHP075_CON_dedupped.bam
CHP078_CON_dedupped.bam
CHP084_CON_dedupped.bam
CHP089_CON_dedupped.bam
CHP102_CON_dedupped.bam
CHP103_CON_dedupped.bam
CHP106_CON_dedupped.bam
CHP107_CON_dedupped.bam
CHP110_CON_dedupped.bam
CHP111_CON_dedupped.bam
CHP112_INF_dedupped.bam
CHP113_INF_dedupped.bam
CHP126_CON_dedupped.bam
CHP127_CON_dedupped.bam
CHP129_CON_dedupped.bam
CHP130_INF_dedupped.bam
CHP132_CON_dedupped.bam
TB__1_trimmed_dedup.bam
TB__13_trimmed_dedup.bam
TB__14_trimmed_dedup.bam
TB__16_trimmed_dedup.bam
TB__17_trimmed_dedup.bam
TB__18_trimmed_dedup.bam
TB__19_trimmed_dedup.bam
TB__20_trimmed_dedup.bam
TB__22_trimmed_dedup.bam
TB__23_trimmed_dedup.bam
TB__24_trimmed_dedup.bam
TB__25_trimmed_dedup.bam
TB__26_trimmed_dedup.bam
TB__3_trimmed_dedup.bam
TB__30_trimmed_dedup.bam
TB__32_trimmed_dedup.bam
TB__34_trimmed_dedup.bam
TB__36_trimmed_dedup.bam
TB__37_trimmed_dedup.bam
TB__38_trimmed_dedup.bam
TB__39_trimmed_dedup.bam
TB__4_trimmed_dedup.bam
TB__41_trimmed_dedup.bam
TB__5_trimmed_dedup.bam
TB__6_trimmed_dedup.bam
```

TB__7_trimmed_dedup.bam

| Genome browser session (e.g. UCSC) | no longer applicable |
|---|---|

## Methodology

| Replicates | Each sample was sequenced once. |
|---|---|

| Sequencing depth | The reads are paired end 2x100 bp. For each sample, the number of non-redundant read-pairs (min 16.8 million read pairs, average 52.6 million read pairs) and the median insert size are given in Supplementary Tables 2, 4, 6, 8 and 19. |
|---|---|

| Antibodies | H3K27ac antibody Catalogue #39133; Active Motif |
|---|---|

| Peak calling parameters | Alignment: bwa mem -t 4 hg19.fa read1.fastq read2.fastq \| samtools view -Sbh - > bamfile.bam

convert bam to bed using bedTools: bamToBed -i bamfile.bam 1>bamtobed.bed

Peak Calling: run_dfilter.sh -d=bamtobed.bed -c=bamtobed_input.bed -o=outputfile -ks=100 -redund=1000 |
|---|---|

| Data quality | Duplicate reads (read-pairs mapping to the same genomic location) were collapsed. For each sample, ChIP-seq peaks were detected using DFilter at a P-value threshold of 1e-6. For either cell type, the initial peak set was defined as the union of peaks from the entire set of discovery and validation samples from Batches 1 and 2, which included the vast majority of samples (Table S1). Peaks wider than 8Kb were then discarded. We performed multi-sample correction for GC bias (refs 13, 37) separately on each processing batch. We then discarded samples with low complexity as measured by the number of unique read-pairs (<15 million for Batches 1 and 2; <30 million for Batch 3). We also discarded samples with high GC bias (>4,500 peaks showing greater than two-fold GC bias in either direction). The number of peaks called for each sample is given in Supplementary Tables 2, 4, 6, and 8. Coordinates (bed format) of DA peaks are in Supplementary Tables 3, 5, 7 and 9. |
|---|---|

| Software | read mapping: bwa 0.7.10-r789, samtools 0.1.19-44428cd
bedtools: bedtools v2.25.0
peak calling: DFilter1.6 |
|---|---|

# Flow Cytometry

## Plots

Confirm that:

☒ The axis labels state the marker and fluorochrome used (e.g. CD4-FITC).

☒ The axis scales are clearly visible. Include numbers along axes only for bottom left plot of group (a 'group' is an analysis of identical markers).

☐ All plots are contour plots with outliers or pseudocolor plots.

☒ A numerical value for number of cells or percentage (with statistics) is provided.

## Methodology

| Sample preparation | PBMCs and granulocytes were isolated from freshly drawn blood. From PBMCs, monocytes were isolated using CD14+ immunomagnetic separation beads (MACS, Miltenyi). Isolated CD14+ and granulocytes were assessed for purity by flowcytometry. In some experiments monocytes were stimulated with potassium for defined time-point. Cells (unstimulated and stimulated) were washed and single-cell suspensions were resuspended in PBS and stained for 30min with live/dead fixable aqua dead cell stain kit (Thermo Fisher Scientific, #L34965). Cells were then stained with respective antibodies. In experiments where phosporylated substrate was assessed, Live/Dead stained cells were washed with FACS buffer (containing PBS with 3% FBS, 1mM EDTA and 0.1% sodium azide) and incubated with Human TruStain FcX™ (BioLegend, #422302) for 20min. Following this, cells were washed with FACS buffer, fixed and permeabilized using the PerFix EXPOSE kit (Beckman Coulter, #B26976) according to manufacturer's instructions. Permeabilized cells were incubated for 30 min in the dark with fluorophore-conjugated antibodies specific for phosphorylated marker or isotype control antibodies. |
|---|---|

| Instrument | BD LSRII |
|---|---|

| Software | BD FACSDiva software was used to collect the acquired flow data. Data was analysed using FlowJo software. |
|---|---|

| Cell population abundance | The purified CD14+ monocytes and granulocytes showed purity >95%. |
|---|---|

| Gating strategy | The CD14+ mononuclear cells were FSC/SSC gated. Upon this single cells were gated using FSC-H/FSC-A. Single cells were used to gate out dead cells using fixable live-dead stain. Live cells were assessed for the staining by antibodies targeting phosphorylated substrates.
Gating strategy for apoptosis experiment has been added in Supplementary Fig 8 |
|---|---|

☒ Tick this box to confirm that a figure exemplifying the gating strategy is provided in the Supplementary Information.

