## [Peer Review File · Nature Microbiology]

Peer Review Information

Journal: Nature Microbiology

Manuscript Title: Histone acetylome-wide associations in immune cells from individuals with active *Mycobacterium tuberculosis* infection

Corresponding author name(s): Shyam Prabhakar

Reviewer Comments & Decisions:

Decision Letter, initial version:

Dear Dr. Prabhakar,

Thank you for submitting your revised manuscript "Histone acetylome-wide association study of tuberculosis" (NMICROBIOL-21092305-T). It has now been seen by the original referees and their comments are below. The reviewers find that the paper has improved in revision, and therefore we'll be happy in principle to publish it in Nature Microbiology, pending minor revisions to satisfy the referees' final requests and to comply with our editorial and formatting guidelines.

Thank you again for your interest in Nature Microbiology Please do not hesitate to contact me if you have any questions.

{redacted}

Reviewer #1 (Remarks to the Author):

The authors have done a thorough job of attending to previously identified comments. They now include an independent longitudinal cohort from South Africa and address statistical issues along with addressing other technical issues. There is now a focus on apoptosis rather than autophagy which is clearer.

Reviewer #2 (Remarks to the Author):

The authors have effectively addressed all my concerns and I have no additional comment.

Reviewer #3 (Remarks to the Author):

2Rosario et al present a significantly revised version of their initial manuscript transferred from Nature Medicine. The authors have taken great care to address all issues raised in my previous review of their work. I appreciate the additional replication analysis, power analysis, TF motif analysis and the many clarifications that the authors provide. Overall, the revision has improved the manuscript substantially. I only have minor remaining comments:

Re comment 5 (replication of haQTL):

I would recommend the authors to include a statement on the use of allelic imbalance (as indicated in their response) also to the main text (between lines 373-388), to help the reader better understand the reasons for the differences in power relative to the sample size between the cohorts / analyses.

Decision Letter, final checks:

Dear Dr. Prabhakar,

Thank you for your patience as we've prepared the guidelines for final submission of your Nature Microbiology manuscript, "Histone acetylome-wide association study of tuberculosis" (NMICROBIOL-21092305-T). Please carefully follow the step-by-step instructions provided in the attached file, and add a response in each row of the table to indicate the changes that you have made. Please also check and comment on any additional marked-up edits we have proposed within the text. Ensuring that each point is addressed will help to ensure that your revised manuscript can be swiftly handed over to our production team.

In recognition of the time and expertise our reviewers provide to Nature Microbiology's editorial process, we would like to formally acknowledge their contribution to the external peer review of your manuscript entitled "Histone acetylome-wide association study of tuberculosis". For those reviewers who give their assent, we will be publishing their names alongside the published article.

Nature Microbiology offers a Transparent Peer Review option for new original research manuscripts submitted after December 1st, 2019. As part of this initiative, we encourage our authors to support increased transparency into the peer review process by agreeing to have the reviewer comments,

2author rebuttal letters, and editorial decision letters published as a Supplementary item. When you submit your final files please clearly state in your cover letter whether or not you would like to participate in this initiative. Please note that failure to state your preference will result in delays in accepting your manuscript for publication.

Cover suggestions

As you prepare your final files we encourage you to consider whether you have any images or illustrations that may be appropriate for use on the cover of Nature Microbiology.

Nature Microbiology has now transitioned to a unified Rights Collection system which will allow our Author Services team to quickly and easily collect the rights and permissions required to publish your work. Approximately 10 days after your paper is formally accepted, you will receive an email in providing you with a link to complete the grant of rights. If your paper is eligible for Open Access, our Author Services team will also be in touch regarding any additional information that may be required to arrange payment for your article.

Please note that Nature Microbiology is a Transformative Journal (TJ). Authors may publish their research with us through the traditional subscription access route or make their paper immediately open access through payment of an article-processing charge (APC). Authors will not be required to make a final decision about access to their article until it has been accepted. Find out more about Transformative Journals

Authors may need to take specific actions to achieve compliance with funder and institutional open access mandates. For submissions from January 2021, if your research is supported by a funder that requires immediate open access (e.g.

3according to [Plan S principles](https://www.springernature.com/gp/open-research/plan-s-compliance)) then you should select the gold OA route, and we will direct you to the compliant route where possible. For authors selecting the subscription publication route our standard licensing terms will need to be accepted, including our [self-archiving policies](https://www.springernature.com/gp/open-research/policies/journal-policies). Those standard licensing terms will supersede any other terms that the author or any third party may assert apply to any version of the manuscript.

Please use the following link for uploading these materials:
{redacted}

{redacted}

Reviewer #1:

Remarks to the Author:

The authors have done a thorough job of attending to previously identified comments. They now include an independent longitudinal cohort from South Africa and address statistical issues along with addressing other technical issues. There is now a focus on apoptosis rather than autophagy which is clearer.

Reviewer #2:

Remarks to the Author:

The authors have effectively addressed all my concerns and I have no additional comment.

Reviewer #3:

Remarks to the Author:

Rosario et al present a significantly revised version of their initial manuscript transferred from Nature Medicine. The authors have taken great care to address all issues raised in my previous review of their work. I appreciate the additional replication analysis, power analysis, TF motif analysis and the many clarifications that the authors provide. Overall, the revision has improved the manuscript substantially.

4I only have minor remaining comments:

Re comment 5 (replication of haQTL):

I would recommend the authors to include a statement on the use of allelic imbalance (as indicated in their response) also to the main text (between lines 373-388), to help the reader better understand the reasons for the differences in power relative to the sample size between the cohorts / analyses.

Author Rebuttal to Initial comments

Author Rebuttal to Initial comments :

Reviewer #1 (Remarks to the Author):

The authors have done a thorough job of attending to previously identified comments. They now include an independent longitudinal cohort from South Africa and address statistical issues along with addressing other technical issues. There is now a focus on apoptosis rather than autophagy which is clearer.

Response: We appreciate the remarks of Reviewer 1.

Reviewer #2 (Remarks to the Author):

The authors have effectively addressed all my concerns and I have no additional comment.

Response: We thank Reviewer 2 for his/her comment.

Reviewer #3 (Remarks to the Author):

Rosario et al present a significantly revised version of their initial manuscript transferred from Nature Medicine. The authors have taken great care to address all issues raised in my previous review of their work. I appreciate the additional replication analysis, power analysis, TF motif analysis and the many clarifications that the authors provide. Overall, the revision has improved the manuscript substantially. I only have minor remaining comments:

Re comment 5 (replication of haQTL):

I would recommend the authors to include a statement on the use of allelic imbalance (as indicated in

5their response) also to the main text (between lines 373-388), to help the reader better understand the reasons for the differences in power relative to the sample size between the cohorts / analyses.

Response: We appreciate the remarks of Reviewer 3. We have now added the following sentences between lines 373-388 of the original submission to Nature Microbiology:

“We also note that our haQTL analysis approach exploited additional information present in the data, namely allelic imbalance in ChIP-seq reads at heterozygous sites. By modeling the effect of genotype on both peak height and allelic imbalance, our approach had relatively high statistical power.”

Final Decision Letter:

Dear Shyam,

I am pleased to accept your Article "Histone acetylome-wide associations in immune cells from individuals with active *Mycobacterium tuberculosis* infection" for publication in Nature Microbiology. Thank you for having chosen to submit your work to us and many congratulations.

6Acceptance of your manuscript is conditional on all authors' agreement with our publication policies (see <https://www.nature.com/nmicrobiol/editorial-policies>). In particular your manuscript must not be published elsewhere and there must be no announcement of the work to any media outlet until the publication date (the day on which it is uploaded onto our website).

Please note that *Nature Microbiology* is a Transformative Journal (TJ). Authors may publish their research with us through the traditional subscription access route or make their paper immediately open access through payment of an article-processing charge (APC). Authors will not be required to make a final decision about access to their article until it has been accepted. [Find out more about Transformative Journals](https://www.springernature.com/gp/open-research/transformative-journals)

Authors may need to take specific actions to achieve compliance with funder and institutional open access mandates. For submissions from January 2021, if your research is supported by a funder that requires immediate open access (e.g. according to [Plan S principles](https://www.springernature.com/gp/open-research/plan-s-compliance)) then you should select the gold OA route, and we will direct you to the compliant route where possible. For authors selecting the subscription publication route our standard licensing terms will need to be accepted, including our [self-archiving policies](https://www.springernature.com/gp/open-research/policies/journal-policies). Those standard licensing terms will supersede any other terms that the author or any third party may assert apply to any version of the manuscript.

You can now use a single sign-on for all your accounts, view the status of all your manuscript

7submissions and reviews, access usage statistics for your published articles and download a record of your refereeing activity for the Nature journals.
